# A pesticide and iPSC dopaminergic neuron screen identifies and classifies Parkinson-relevant pesticides

Kimberly C. Paul [1,14] ✉, Richard C. Krolewski [2,3,14], Edinson Lucumi Moreno[2], Jack Blank [11], Kristina M. Holton[3], Tim Ahfeldt[12,13], Melissa Furlong[4], Yu Yu [5], Myles Cockburn[6], Laura K. Thompson [6], Alexander Kreymerman[3], Elisabeth M. Ricci-Blair[3], Yu Jun Li[3], Heer B. Patel[3], Richard T. Lee [3,7,8], Jeff Bronstein[1], Lee L. Rubin [3,7,15] ✉, Vikram Khurana [2,9,15] ✉ & Beate Ritz [1,10,15] ✉

Parkinson's disease (PD) is a complex neurodegenerative disease with etiology rooted in genetic vulnerability and environmental factors. Here we combine quantitative epidemiologic study of pesticide exposures and PD with toxicity screening in dopaminergic neurons derived from PD patient induced pluripotent stem cells (iPSCs) to identify Parkinson's-relevant pesticides. Agricultural records enable investigation of 288 specific pesticides and PD risk in a comprehensive, pesticide-wide association study. We associate long-term exposure to 53 pesticides with PD and identify co-exposure profiles. We then employ a live-cell imaging screening paradigm exposing dopaminergic neurons to 39 PD-associated pesticides. We find that 10 pesticides are directly toxic to these neurons. Further, we analyze pesticides typically used in combinations in cotton farming, demonstrating that co-exposures result in greater toxicity than any single pesticide. We find trifluralin is a driver of toxicity to dopaminergic neurons and leads to mitochondrial dysfunction. Our paradigm may prove useful to mechanistically dissect pesticide exposures implicated in PD risk and guide agricultural policy.

Parkinson's disease (PD) is a complex, multi-factorial neurodegenerative disease. The hallmark pathology of PD is aggregation of the protein α-synuclein in Lewy bodies in specific midbrain dopaminergic (mDA) neurons. Etiologic contributors include genetic, environmental factors, and aging[1]. Ample evidence links pesticides in general to PD etiology[2]. In California, which is the largest agricultural producer and exporter in the United States, there are currently 13,092 pesticide products with 1059 different active ingredients registered for use[3]. While pesticides

[1]Department of Neurology, UCLA David Geffen School of Medicine, Los Angeles, CA, USA. [2]Division of Movement Disorders, Department of Neurology, Brigham and Women's Hospital and Harvard Medical School, Boston, MA 02115, USA. [3]Department of Stem Cell and Regenerative Biology, Harvard University, Cambridge, MA, USA. [4]University of Arizona, Mel and Enid Zuckerman College of Public Health, Tucson, AZ, USA. [5]UCLA Center for Health Policy Research, Los Angeles, CA, USA. [6]Department of Population and Public Health Sciences, Keck School of Medicine, University of Southern California, Los Angeles, CA, USA. [7]Harvard Stem Cell Institute, Cambridge, MA, USA. [8]Division of Cardiovascular Medicine, Department of Medicine, Brigham and Women's Hospital and Harvard Medical School, 75 Francis St, Boston, MA 02115, USA. [9]Broad Institute of MIT and Harvard, Cambridge, MA 02142, USA. [10]Department of Epidemiology, UCLA Fielding School of Public Health, Los Angeles, CA, USA. [11]Present address: Prime Medicine, Inc, Cambridge, MA, USA. [12]Present address: Recursion Pharmaceuticals, Salt Lake City, UT, USA. [13]Present address: Nash Family Department of Neuroscience at Mount Sinai, New York, NY, USA. [14]These authors contributed equally: Kimberly C. Paul, Richard C. Krolewski. [15]These authors jointly supervised this work: Lee L. Rubin, Vikram Khurana, Beate Ritz. ✉e-mail: kimberlp@ucla.edu; lee_rubin@harvard.edu; vkhurana@bwh.harvard.edu; britz@ucla.edu

are important components of modern commercial agriculture that help maximize food production, most pesticides that are applied at industrial scales have not been adequately assessed for their potential role in PD, let alone for their mechanisms of action. Aside from important work in model organisms and cellular models mostly focused on rotenone and paraquat, the specific effects of most common, widely used pesticides remain unexplored[4]. Fewer studies still have delved into the effect of co-exposures and whether pesticides may directly exert effects on human dopaminergic neurons that are particularly sensitive to environmental toxicants through oxidative stress[5–9]. A wider screen of commercially applied pesticides in relation to PD, coupled to analysis in tractable human dopaminergic neuron models, may thus identify new disease targets, provide mechanistic insights, and help revise priorities for research and public health policy.

Responding to this need, here we have developed a field-to-bench paradigm, coupling systematic epidemiologic screening with direct testing in neurons to assess and mechanistically dissect pesticide-PD relationships. First, we performed a pesticide-wide association study (PWAS). We established a record-based exposure assessment approach using agricultural pesticide application records in California to comprehensively investigate long-term ambient pesticide exposure in relation to PD risk. This enabled agnostic screening of nearly 300 specific pesticide active ingredients in an untargeted manner, without relying on self-reported exposure or pre-selection of specific pesticides. We followed this with a systematic analysis of the effects of pesticide hits on dopaminergic (mDA) neurons derived from PD patient-induced pluripotent stem cells (iPSC). This cellular platform enabled us to directly test whether pesticides identified via our PWAS exert an adverse effect on PD-patient derived mDA neurons.

We chose mDA neurons produced from iPSCs as our model-system because they represent an excellent tool for personalized in vitro disease modeling, including for central nervous system cells[10–13]. Moreover, human iPSC model systems express proteins at endogenous levels and harbor disease-relevant pathologies, including mitochondrial function, ER-to-Golgi trafficking, reduced protein synthesis, increased nitrosative stress, and deficient survival over time[8,10–17]. In this study, we used iPSCs from two different PD patients expressing wild-type α-synuclein, either at endogenous levels or increased levels (from triplication) known to drive aggressive early-onset PD. We differentiated these iPSCs into mDA neurons, a key neuron affected in PD and known to be highly sensitive to oxidative stress. Importantly, human mDA neurons exhibit fundamental differences from rodent or other human cell lines, most dramatically with the biology of dopamine oxidation[6]. While protocols for differentiation of iPSC into midbrain dopaminergic neurons have steadily improved, heterogeneity of differentiation line-to-line, clone-to-clone, and experiment-to-experiment remains a challenge. To overcome this, we recently developed a bright red fluorescent reporter engineered into the tyrosine hydroxylase locus, enabling us in this study to specifically assess the effects of pesticides on mDA neurons, free from the presence of other cell types[14].

Here we show that among the multitude of potentially PD-relevant pesticides we identified, we were able to pinpoint ten that were directly toxic to mDA neurons. Data on co-exposures, common in agricultural practices, allowed us to develop co-exposure paradigms "in the dish" to test whether combinations of pesticides lead to greater, synergistic toxicity. For example, from pesticides used in combination in cotton agriculture, we identified trifluralin together with other commonly co-applied pesticides as being significantly more toxic to mDA neurons than any of the cotton-applied pesticides alone. We attributed the trifluralin-driven neurotoxicity to mitochondrial dysfunction in those neurons. In time, this approach will enable us to further track such cellular pathologies back to epidemiologic and environmental data, to mechanistically understand the individual and combined effects of pesticides, and to hopefully help inform the judicious use of pesticides in agriculture.

## Results

### Population-based study overview

Since 1972, California law mandates the recording of commercial pesticide use to the pesticide use report (PUR) database, documenting nearly 50 years of agricultural application of hundreds of pesticides. We have designed a geospatial algorithm which combines this database with maps of land-use and crop cover to determine for each individual pesticide active ingredient in the PUR, the reported pounds of pesticide applied per acre within a 500 m buffer around specific locations, such as addresses, yearly since 1974[15]. We applied this system to lifetime residential and workplace address histories from participants of the Parkinson's Environment and Genes (PEG) study ($n = 829$ PD patients and $n = 824$ controls recruited as part of two independent study waves; see Methods).

PEG is a population-based Parkinson's Disease case-control study conducted in three agricultural counties in Central California[16]. Patients were enrolled early in their disease course and all were seen by UCLA movement disorder specialists for in-person neurologic exams and confirmed as having clinically-defined, idiopathic PD[17]. For each pesticide active ingredient, hereafter referred to as "pesticide", in the PUR and each PEG participant, we determined the average pounds of pesticide applied per acre per year within a 500 m buffer of each residential and workplace address over the study window (1974 to 10 years prior to index date, which was PD diagnosis for patients or interview date for controls). This approach created one summary estimate of the average pounds of pesticide applied per acre per year within the 500 m buffer for each PUR pesticide.

### Description of agricultural pesticide applications in the study area, including range and location of exposure

From 1974 to 2017, there were approximately 5.9 million PUR records in the tri-county study area, documenting the application of 1355 unique pesticides. Figure 1a details the study region and all agricultural pesticide applications reported in 2000, when PEG began. The number of different PUR-reported pesticides applied per year across the three counties, which ranged from a low of 288 in 1977 through a high of 646 in 2005, and the total reported pounds applied per year aggregated across pesticides, which peaked in 1998, are shown in Fig. 1b. Of the 1355 different pesticides applied from 1974–2017, 722 were applied within the 500 m buffer of at least one PEG participant's residence or workplace.

On average, the PD patients in the study both lived and worked near commercial agricultural facilities applying more total pounds of pesticide per acre (Fig. 1c) than controls (average annual mean difference: 133 more pounds of pesticide applied per acre per year near the patients' residences versus controls' and 343 more pounds near workplaces). Of the 722 different active ingredients applied within the study participants' buffer zone, PD patients and controls on average lived near the application of 50 (SD = 44.4) and 45 (SD = 40.9) different pesticides, respectively, during the entire exposure window. The mean number near participants' workplace was 50 (SD = 45.4) for patients and 38 (SD = 39.2) for controls. The number of pesticides by year is shown in Supplementary Figure 1.

Similar differences were observed in each study wave independently, for men and women separately, and when limiting to the 288 pesticides with ≥25 exposed participants (Supplementary Data 1). Figures displaying the median values are shown in Supplementary Figure 2.

### Individual pesticide associations with PD in a pesticide-wide association analysis (PWAS) described according to pesticide class, use type, and an overrepresentation analysis

We assessed each pesticide individually for PD risk in what we call a PWAS. Of the 722 pesticides described above, we included 288 in our PWAS, according to our criterion of having ≥25 exposed study

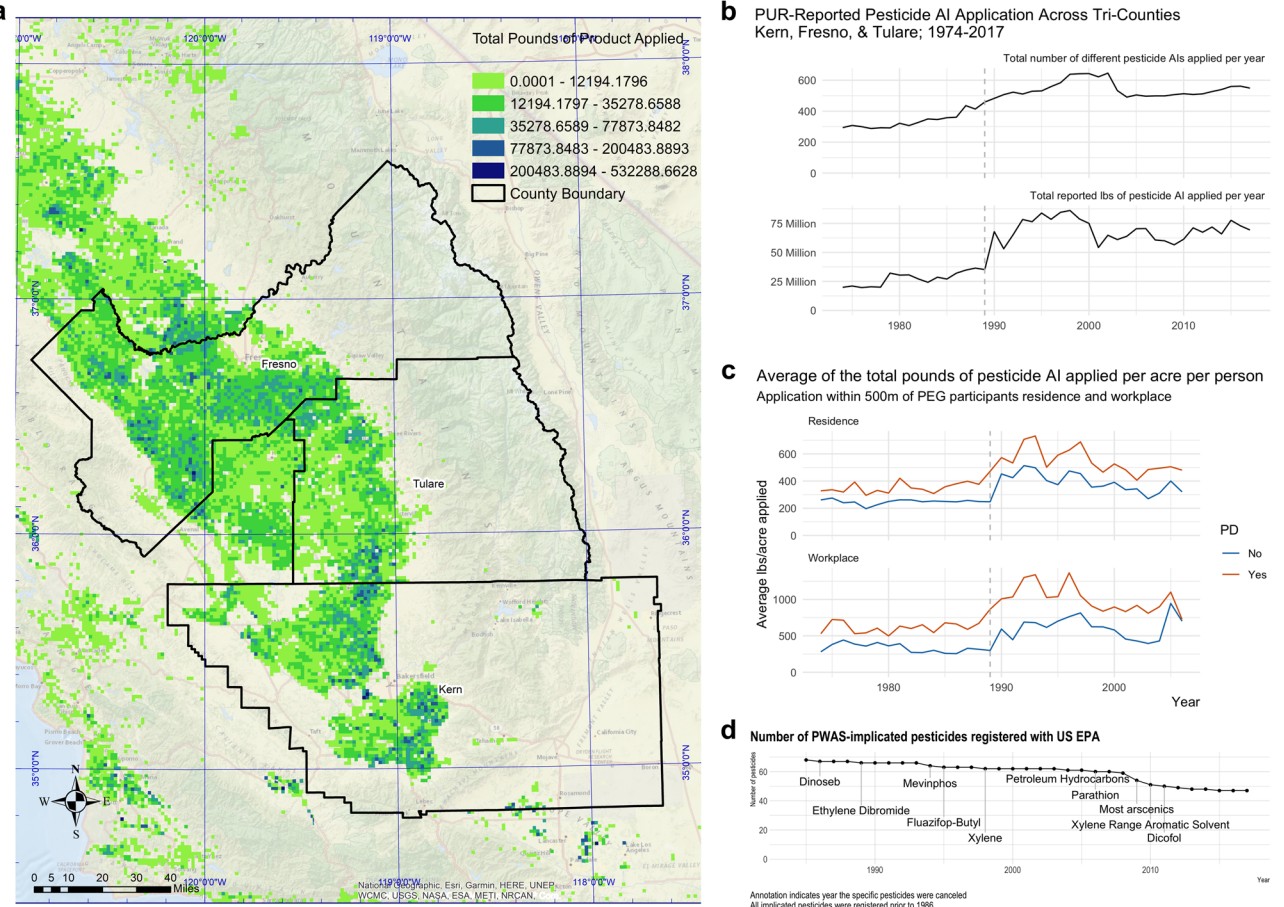

**Fig. 1 | Description of agricultural pesticide use in the study area, including geography of applications, number of unique active ingredients applied by year, total pounds applied, and pesticide registration timeline. a** Geography of study region for PEG cohort and total pounds of pesticides applied in the region in 2000. Total pounds of pesticides applied shown by color scale. **b** The number of different PUR-reported pesticides applied per year across the three counties and the total reported pounds of pesticide applied per year across the three counties (1974–2017). **c** The average total reported pounds of pesticide applied per acre around PEG participants' residential and workplace addresses per year from 1974–2006 (the mean index year), by PD status. Values above the 99th percentile were limited to the 99th percentile. **d** Timeline showing the number of PWAS-implicated pesticides that were registered with the US EPA by year. The annotation indicates the year the named pesticide had registration canceled or withdrawn. Source data are provided as a Source Data file.

participants. Due to special considerations for paraquat dichloride, specifically strong experimental support for the hypothesis and the interest in estimating the effects of duration and intensity of exposure, we present results from these analyses in a separate manuscript[18].

Figure 2a shows a Manhattan plot delineating the statistical significance for each pesticide, grouped by use type. Our PWAS implicated 25 pesticides as associated with PD at a meta-analysis FDR ≤ 0.01 (8.7% of all tested pesticides), another 28 at 0.01<FDR ≤ 0.05 (9.7%), and 15 at 0.05<FDR < 0.10 (5.2%) (Fig. 2b). The top five associated pesticides by FDR were sodium chlorate, dicofol, prometryn, methomyl, and xylene range aromatic solvent. These pesticides showed consistent risk profiles across location and study waves (Supplementary Fig. 3). Exposure descriptive statistics and risk estimates stratified by study wave and exposure location for pesticides associated with PD at $p < 0.10$ can be found in Supplementary Data 2 and 3.

Regulatory and toxicity information for the implicated pesticides is shown in Supplementary Data 4. Eighteen of the 25 most strongly PD-associated pesticides (FDR ≤ 0.01) are actively registered with the US EPA (43 of 68 pesticides at FDR < 0.10, Fig. 1d), while only 2 are allowed for use in the EU at time of publication. Of these 25 pesticides, 21 are considered 'bad actors' by the Pesticide Action Network (PAN)[19] as 9 have been deemed carcinogens (7 more as possible carcinogens), 6 developmental or reproductive toxins, 10 cholinesterase inhibitors, 3 known groundwater contaminants (13 more as possible groundwater

contaminants), and 8 have high acute toxicity. In fact, of the 53 pesticides with an FDR ≤ 0.05, 43 have been designated 'bad actors'.

We used overrepresentation analysis (ORA) to test for enrichment of pesticide groups (toxicity groups, chemical classes, and use types) in the PD-associated pesticide set (PWAS FDR ≤ 0.05, $n = 53$ pesticides) relative to all pesticides assessed ($n = 286$ pesticides with group classifications; Supplementary Data 5). The ORA, which is commonly applied to evaluate gene-set overrepresentation, allowed us to identify classes or types of pesticides that were associated with PD more than expected and thus potentially have group characteristics important in PD etiopathogenesis.

The ORA indicated that seven of the toxicity classifications – including cholinesterase inhibitors, highly hazardous pesticides, acutely toxic pesticides, and carcinogens – were overrepresented (Fig. 2c). This indicates there were more PD-associated pesticides with these toxicity classifications than expected based on the group's distributions among all assessed pesticides. Furthermore, the odds of being among the PD-associated pesticides was also 2 to 3-fold higher for pesticides that contaminate groundwater and for those that are highly prone to drift. Figure 2d shows the percent of pesticides in each group that were associated with PD, the numbers are detailed in Supplementary Data 5.

Several pesticide chemical classes and use types were also significantly overrepresented, though small numbers limited statistical

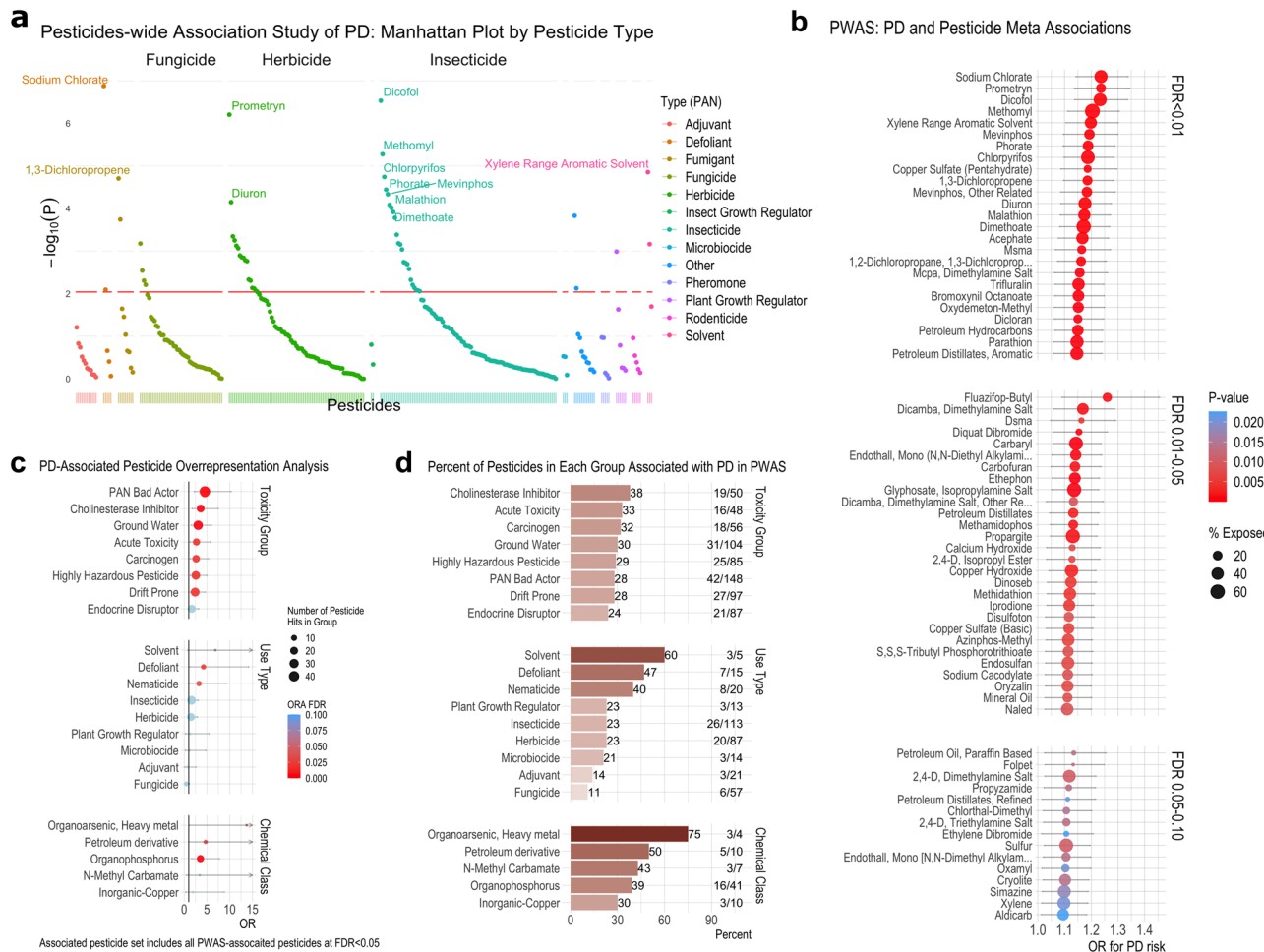

**Fig. 2 | Pesticide-Wide Association Study analysis associated specific pesticides with PD and overrepresentation analysis implicates groups of pesticides overrepresented in the associated pesticides. a** Manhattan plot detailing the -log(p-value) from the meta-analysis for all 288 pesticides tested for association with PD. We conducted univariate, unconditional logistic regression to calculate odds ratios (ORs) and 95% confidence intervals (CIs) for PD with each pesticide (n = 288). We combined the OR estimates from each study wave and location (residential and occupational addresses) in a fixed effects meta-analysis, results shown here. P-values were based on a z-score statistic and two-sided interval. P-values were adjusted for multiple testing using an FDR and are shown in Supplementary Data 3. The red horizontal line indicates the FDR = 0.05 cut-off. **b** Dot plot displaying the odds ratio (OR; dot) and 95% CI (error bars) from the meta-analysis described above for all pesticides with an FDR < 0.10. Analysis for Figs. 2a and 2b was based on n = 829 PD patients and n = 824 controls. The log odds ratio is the center of the 95% CI on the logarithmic scale. The log odds ratio and 95% CI on the logarithmic scale were exponentiated to get the odds ratio and 95% CI. **c** Results of overrepresentation analysis to test for overrepresentation of pesticide groups (toxicity groups, chemical classes, and use types) in the set of PWAS PD-associated

pesticides relative to all pesticides we assessed. Odds ratios (dot) and 95% CIs (error bars) are displayed. The log odds ratio is the center of the 95% CI on the logarithmic scale. The log odds ratio and 95% CI on the logarithmic scale were exponentiated to get the odds ratio. Given the asymmetrical nature of the resulting odds ratio, the odds ratio is no longer the center of the 95% CI. The overrepresentation analysis was based on n = 286 pesticide associations. The associated pesticide set includes all associated pesticides at FDR < 0.05 (n = 53 pesticides). **d** Bar graph indicating the percent of pesticides in each group associated with PD in the PWAS. The graph also shows the total number of pesticides tested in the PWAS from each group (denominator) and the number of pesticides in each group associated with PD (numerator) on the right. This information is used for the overrepresentation analysis. For example, there were 50 cholinesterase inhibitor pesticides assessed for association with PD, 17% of all tested pesticides (50/286). In total, 19 cholinesterase inhibitors were associated with PD at FDR < 0.05 in the PWAS (19/50, 38%). Using an odds ratio and Fisher's exact test, we found that the odds of being among the PD-associated pesticides was 3.6-fold higher for the cholinesterase inhibitors versus the non-cholinesterase inhibiting pesticides (OR = 3.62, 95% CI = 1.73. 7.50, FDR = 3.2e-03). Source data are provided as a Source Data file.

power. These included the defoliant, nematicide, and solvent use types and organophosphorus, heavy metal organoarsenic, and petroleum-derivative chemical classes. For example, while only four heavy metal organoarsenic pesticides were tested for association with PD in our PWAS, three (or 75%) were associated with PD (see Fig. 2d).

Finally, we also preformed several sensitivity analyses. The PWAS results were generally robust to including an indicator for occupational use of pesticides or fertilizer in the model (results shown in Supplementary Data 6) and including an additional set of controls (described in the supplementary materials and shown in Supplementary Data 7). Results stratified by gender are shown in Supplementary Fig. 4. The associations were mostly similar between men and women,

although several pesticides showed stronger associations among men. We tested this by including an interaction term (pesticide*gender) in the models for the 68 pesticides associated with at FDR < 0.10. Five pesticides showed statistically significant interactions (interaction p < 0.05; Supplementary Data 8). However, further investigation is needed to validate and interpret these results, as none of the interactions were statistically significant after multiple testing correction.

## Multiple PWAS-identified pesticides are toxic to iPSC-derived dopaminergic neurons from a PD patient

We tested toxicity of the PWAS-associated pesticides directly in iPSC-derived mDA neurons derived from a patient with PD who harbored

a pathologic triplication at the α-synuclein-encoding *SNCA* locus[20]. This line, from a male member of the Iowa kindred, was selected because it over-expresses wild-type α-synuclein and reflects an extreme form of the pathology associated with idiopathic PD, namely the accumulation of wild-type α-synuclein in neurons. Neurons derived from this and similar SNCA triplication iPSC lines exhibit PD-relevant phenotypes[20–25]. We thus considered it a line "sensitized" to PD-relevant stressors. We engineered these iPSCs to express a fluorescent reporter at the tyrosine hydroxylase (TH) locus to overcome the heterogeneity of mDA differentiation. This THtdTomato reporter was targeted via CRISPR-Cas9 and confirmed via PCR and Sanger sequencing (Fig. 3a–c)[14]. Endogenous THtdTomato signal colocalized with anti-TH-labelling and was consistent with the expression pattern observed in mDA neurons derived from embryonic stem cell reporter lines (Fig. 3f)[14]. This reagent enabled us to selectively evaluate the effects of pesticides on mDA neurons and exclude other contaminating cell types present in patterned iPSC-derived cultures.

Thirty-nine of the pesticides identified in the above PWAS analysis were solubilized in DMSO, water, or ethanol. A four-point dose curve protocol was used spanning a range of pesticide concentrations comparable to previously published toxicity assays performed on human cell lines including neural lineage cells[26,27]. THtdTomato-positive neurons were purified by fluorescence activated cell sorting (FACS) (Supplementary Fig. 5 shows gating strategy; Supplementary Fig. 6 provides assay overview) and used to test for sensitivity to the pesticides in a live-imaging survival assay. Quantitation of the raw number of THtdTomato-positive mDA neurons was performed at baseline prior to treatment, at seven days after treatment, and at eleven days after initial treatment (Supplementary Fig. 6 and Supplementary Fig. 7a, c). Eleven days after initial treatment was chosen as the primary end point for the assay to allow for detection of pesticides that cause rapid cell death and those that cause cell death over a more prolonged assay timeline to be detected in the same assay. At the eleven-day post-treatment endpoint, the Z-prime score for DMSO (negative control) and rotenone (positive control) was 0.549. Ziram, a known mDA neuron toxicant[28], served as an additional positive control. Rotenone produces cell death at eleven days but not seven days (Supplementary Fig. 7c). Ziram produces cell death more rapidly.

Ten PWAS pesticides led to cell death >3 standard deviations above the DMSO control mean at a concentration of 30 μM: propargite, copper sulfate (basic and pentahydrate), dicofol, folpet, naled, endothall, trifluralin, endosulfan, and diquat dibromide (Fig. 4). For example, propargite treatment produced extensive mDA neuron death and degeneration of neurites (Fig. 4b *upper* and *lower*). Dose-response curves extending below the screening concentration indicated that propargite was the most toxic, followed by diquat dibromide, folpet, and naled. All were toxic at 6 μM as well (Fig. 4c). The large standard deviation seen for dicofol was due to an outlier well. Repeat testing confirmed toxicity at concentrations of 9.5 μM and greater (Supplementary Fig. 8). Information about pesticide use type, chemical class, regulation status, and prior toxicity classifications for the 10 pesticides which produced substantial cell death is included in Supplementary Data 9. These compounds encompass a range of use types (four insecticides, three herbicides, and two fungicides), are structurally distinct, and do not have an overlapping prior toxicity classification, such as acute toxicity. Orthogonal imaging-based measurements of cell health, including neurite length, cell area, and pixel intensity of THtdT reporter were measured independent of cell count to confirm toxicity (Supplementary Fig. 7d, e).

The toxicity data determined by cell counts presented in Fig. 4 relies upon fluorescence from the THtdT knock in construct. A pesticide that dramatically reduced expression of the TH gene or greatly exacerbated degradation of the THtdTomato reporter without causing cell death could produce a false positive in this experiment. We thus performed an orthogonal viability assay using CellTitreGlo (Promega) according to the same experimental conditions as in Fig. 4 on a subset of pesticides identified as toxic in the initial screen. Viability decreased as expected with concentrations at or below the original screening concentration (Supplementary Fig. 9, $n = 1$). While we focused on the SNCA triplication line, it is possible that the increased α-synuclein levels in that line (known to cause early onset aggressive PD) may not be reflective of neurons expressing wild-type levels of α-synuclein. We thus generated an additional line with a THtdTomato reporter. This line is derived from a PD patient harboring a point mutation in the SNCA gene, which was then genetically corrected. Alpha-synuclein levels are thus endogenous with 2-copies of the wild-type gene and yet the genetic background is "permissive" to PD. All pesticides that were toxic to the SNCA triplication iPSC line also caused cell death at the screening concentration or lower in these mDA neurons (Supplementary Fig. 10).

We performed a second control to assess specificity. At our screening concentrations, our top-hit pesticides could theoretically be toxic to any cell type. One advantage of the iPSC system is that it can be differentiated to multiple cell types. Here, we selected cardiomyocytes to address this question given their established use in evaluating off-target effects of candidate drugs[29,30]. Cardiomyocytes were generated from the same SNCA triplication iPSC line. We focused on cell count at 11 days post-exposure, just as we had in mDA neurons. Cardiomyocyte beating and purity was confirmed prior to treatment with pesticides. Using the same assay timeline as the neuronal survival assay, cardiomyocytes were counted at eleven days after the first treatment with pesticides (Supplementary Fig. 11). Three of the ten pesticides (dicofol, folpet, and naled) resulted in a statistically significant reduction in cell counts ($n = 2$), while two others (diquat and propargite) trended towards reduction. A lower concentration of propargite was tested than in the original screen due to the high potency of this pesticide. Five of the ten pesticides that were toxic to mDA neurons did not demonstrate substantial cardiomyocyte toxicity by cell count at concentrations that produced significant cell death of mDA neurons: trifluralin, endothall, endosulfan, copper sulfate basic, and copper sulfate pentahydrate. These data suggest a relative selectivity for neurons over cardiomyocytes at the assayed doses.

## PWAS-identified pesticides cluster based on correlation of exposures

Combinations of different pesticides are regularly applied to the same fields within the same season, year after year. We thus investigated how exposures among the PWAS-implicated pesticides were correlated. Further, we asked how the pesticides directly toxic to mDA neurons related to other PWAS-implicated pesticides, either those that did not produce significant mDA death or those that could not be tested. Finally, we used real-world exposure clusters to motivate combinatorial assessment of pesticides, and potential synergistic toxicity, in our mDA neuron assay.

Figure 5a details the correlation between all PWAS-implicated pesticides (FDR < 0.10, $n = 68$ pesticides). The heatmap, which shows broad patterns of correlation, indicates multiple groups of highly correlated pesticides. The complete pairwise correlation tables are in Supplementary Data 10 and 11. To assess how the mDA toxic pesticides correlated with the other PWAS-implicated pesticides, we used a data-driven integration and network analysis approach to assess correlations across two layers: the set of mDA toxic pesticides and the set of all other PWAS-implicated pesticides (Fig. 5b). All correlations between layers at R > 0.45 are shown in the circle and a detailed description of the figure is provided in the legend. An alternative layout of the network showcasing clustered pesticides is shown in Supplementary Fig. 12.

Overall, exposure to the mDA-neurotoxic pesticides was highly correlated with exposure to other PD PWAS-associated pesticides. Specifically, 75% of the PD-associated pesticides from the PWAS that

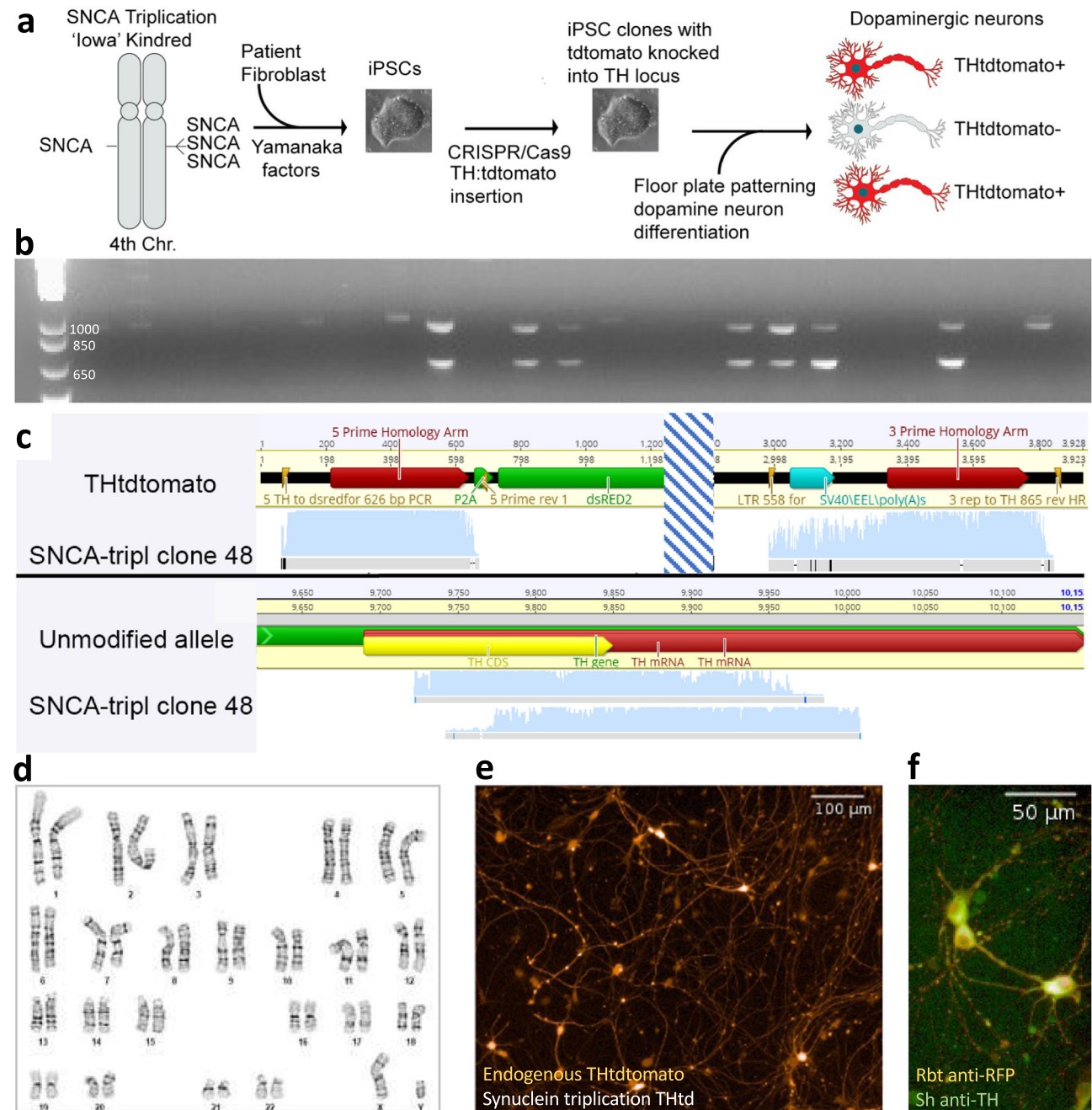

**Fig. 3 | SNCA triplication THtdTomato reporter generation and quality control to facilitate purification and live-cell imaging of dopaminergic neurons.**
**a** Schematic demonstrating iPSC source, generation, modification, and differentiation with tdTomato reporter permitting identification and isolation of dopaminergic neurons. Previously described iPSC line derived from a patient with Parkinson's Disease caused by triplication of the SNCA locus resulting in four copies of the gene encoding α-synuclein. iPSC line was then modified with tyrosine hydroxylase:tdTomato reporter[14]. Adapted from Hallaci et al. 2022[25]. **b–f** Quality control and validation of SNCA triplication THtdTomato reporter line including 5′ and 3′ PCR products to confirm proper insertion. **b** Agarose gel stained with ethidium bromide to demonstrate examples of seven clones that contain the expected PCR products (626 bp product confirmed proper insertion of the 5′ end of the reporter construct and an 878 bp product confirming proper insertion at the 3′ end). PCR reactions run separately but combined into the same wells of the agarose gel for each clone to visualize clones passing and failing PCR quality control.

A subset of clones have a single larger band and these are excluded from further testing. Band length was reproduced in an additional PCR from clones showing proper size to evaluate by Sanger sequencing. **c** Sanger sequencing of PCR products in (**b**) confirming correct insertion of tdTomato cassette. **d** G-banded karyotype (performed by WiCell) confirms normal karyotype in modified clone. **e** Example of live imaging of endogenous THtdTomato fluorescence at 10x. Neurons with this live imaging morphology and appearance are consistently obtained from multiple differentiations (greater than ten) from this cell line. **f** Immunofluorescence co-localization of Rabbit (Rbt) anti-RFP and Sheep (Sh) anti-tyrosine hydroxylase visualized with Alexa Fluor 546 donkey anti-rabbit and Alexa Fluor 488 donkey anti-sheep, respectively. Colocalization of anti-RFP and anti-tyrosine hydroxylase staining reproduces in greater than three differentiations in the cell line used for these experiments. Source data are provided as a Source Data file.

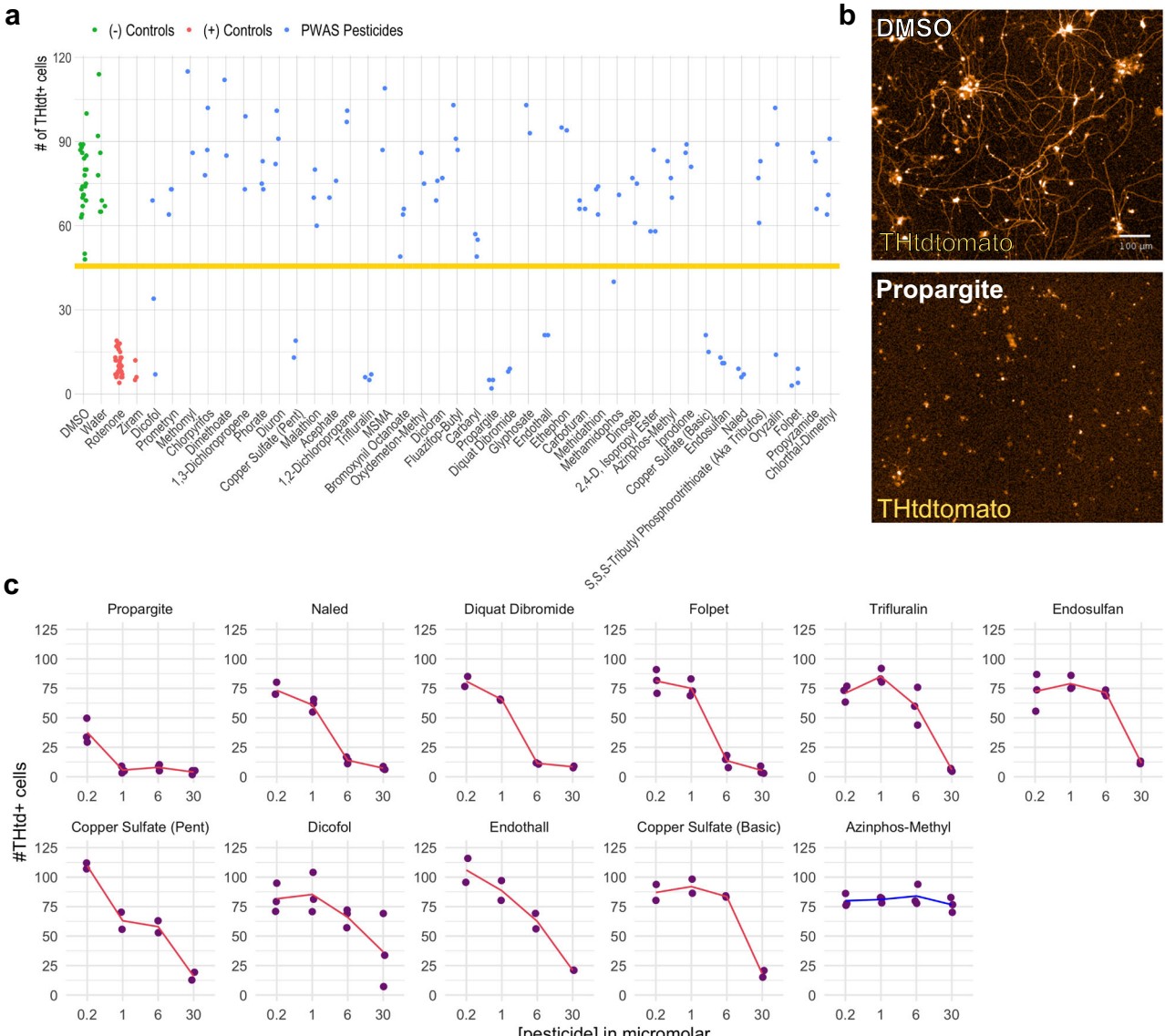

**Fig. 4 | PWAS pesticides are directly toxic to PD patient-derived mDA neurons in a live imaging screen. a** Scatter plot with the number of THtdTomato+ cells measured by live imaging analysis 11 days after the first treatment. DMSO controls (green data points) were present on each assay plate. Water control (green data points) was present on the assay plate containing primarily water-soluble pesticides. Rotenone (red data points) and ziram (red data points) were used as positive controls. Blue data points represent the different pesticides from the PWAS study. Horizontal line denotes three standard deviations below DMSO mean. **b** Upper image is a x10 magnification live image of a DMSO control well. Lower image is from a propargite treated well. Scale bar = 100 μM. More than five independent experiments repeated with DMSO and propargite treatment showed similar neuronal morphology with DMSO and similar extent of cell death and debris with propargite. **c** Four concentration dose curves of PWAS toxicants producing death in SNCA triplication THtdTomato sorted neurons. Cell numbers measured by high-content imaging of live cultures 11 days after first treatment. *n* = 1 with two technical replicates per water soluble pesticide and three technical replicates per DMSO and ethanol soluble pesticides per dose per pesticide for the screen dose curves. Red lines connect average cell number value for each pesticide concentration. Source data are provided as a Source Data file.

either did not result in significant mDA cell death or could not be tested were correlated at R ≥ 0.45 with at least one pesticide that did result in significant mDA cell death. Put another way, exposure to most of the pesticides tied to PD in our PWAS, including those that did not directly lead to mDA neuron toxicity in our assay, was highly correlated with exposure to a pesticide that did exhibit such toxicity. Dicofol emerged as the strongest network hub, being the most interconnected with the highest closeness centrality and number of edges, followed by propargite and methomyl.

Next, we defined correlation clusters representing real-world co-exposure relying on an unsupervised, hierarchical clustering approach (Fig. 6a). In total, we detected 25 different pesticide clusters using residential-based exposure. These are shown as different colors in the dendrogram (Fig. 6a) and detailed in Supplementary Data 12 and 13. Some of the clusters consisted of a single pesticide, meaning these pesticides did not correlate above 0.45 with any other pesticide. The other clusters represent co-exposure profiles of interest for further in vitro testing.

We isolated one cluster of particular interest with clear co-application patterns that also contained several of the most statistically significant PD PWAS-pesticides (Fig. 6b, Cluster 7 in the Supplementary Data). Half of this cluster were defoliants, a unique type of pesticide which causes leaves to fall off plants. Defoliants are applied in agriculture almost exclusively on cotton. We confirmed this by aggregating all PUR records in the tri-county area over the study period and assessing the proportion of the applications for

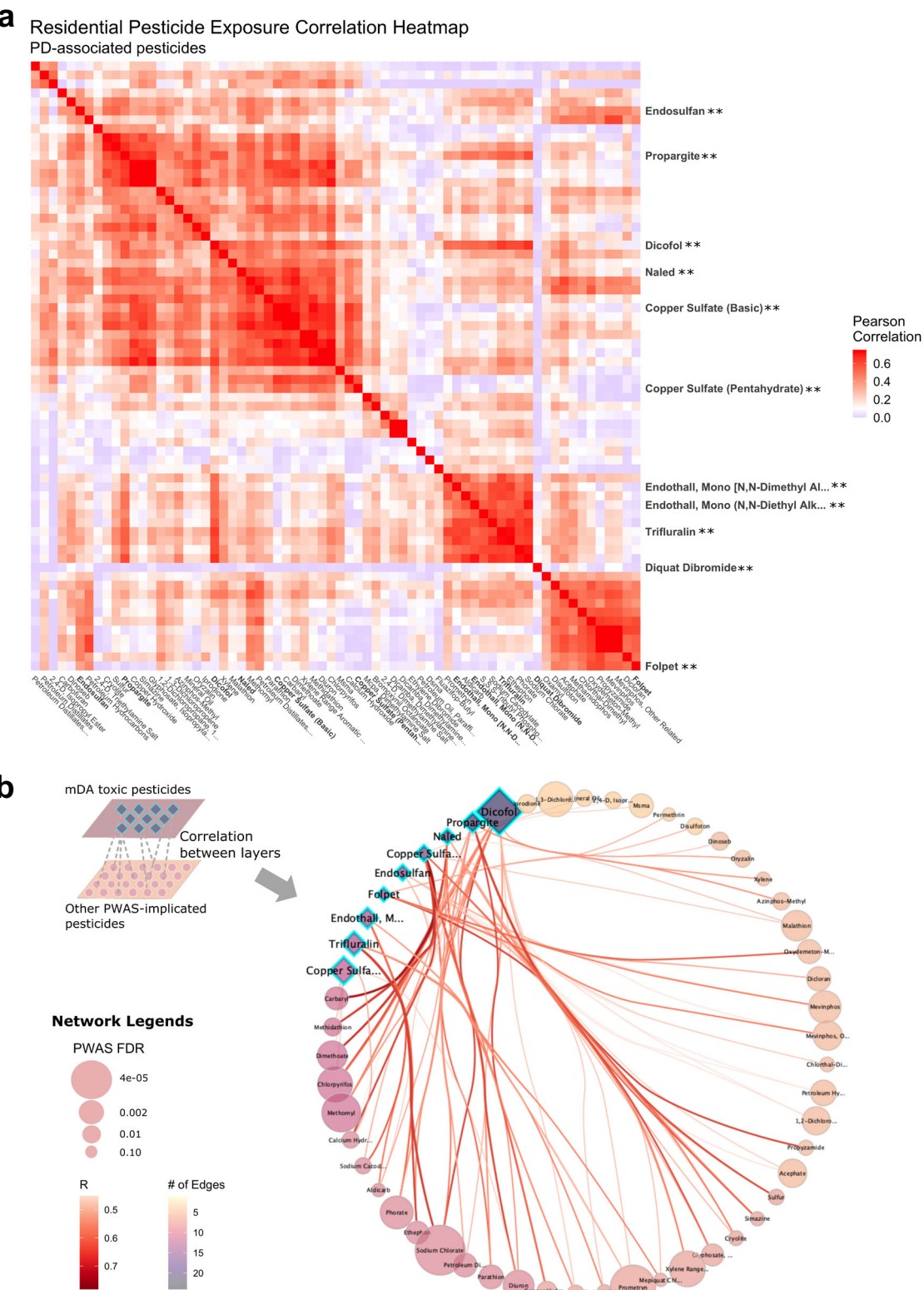

each pesticide on cotton (Fig. 6b). We found 99.96% of the reported S,S,S-tributyl phosphorotrithioate (tribufos) applications were on cotton, 99.76% of sodium cacodylate, and 99.6% of sodium chlorate. Given the clear patterns of real-world co-exposure due to proximity to cotton agriculture, strength of the individual PWAS associations, and manageable cluster size for in vitro combinatorial testing, we selected members of this "cotton cluster" for further co-exposure experiments.

### In vitro co-exposure testing identifies pesticides with most potent direct effects on dopaminergic neurons from the cotton cluster

Co-exposures can be modeled in vitro via concurrent treatment to directly compare the consequences of multiple direct co-exposures. We hypothesized that pesticides with the most relevance to PD risk would be likely to cause mDA neuron cell death on their own, but that toxicity could be enhanced by co-exposure to other pesticides

**Fig. 5 | Pesticide exposure correlations demonstrate substantial interconnections of pesticides that produced significant mDA neuron death with many other pesticides. a** Correlation heatmap indicating the pairwise Pearson correlation coefficient for 68 PWAS-implicated pesticides (FDR < 0.10), using residential address-based exposures. The pesticides which produced significant mDA neuron death in the iPSC-model are highlighted on the y-axis. No pesticides were significantly negatively correlated. Thus, the blue color represents null (R = 0) correlation to dark red representing strong correlation (R > 0.75). All pesticide labels are shown on the x-axis, the y-axis only displays labels of select pesticides, with the ** indicating that the pesticides were toxic to mDA neurons. **b** Correlation wheel showing the pesticide exposure correlations across two layers: first, the set of mDA toxic pesticides, which are designated as teal highlighted diamonds, and second, the set of all other PWAS-implicated pesticides, shown as circles. Correlations between layers at R > 0.45 are shown in the circle, correlations within layers are not shown. The size of the shapes in the correlation circle (diamonds and circles) were

determined by the PWAS FDR, thus pesticides that were more strongly associated with PD in the PWAS are represented by larger sized shapes. The color of the shapes reflects the density of the connections (i.e. correlations at R > 0.45) made by that specific pesticide with others. Pesticides with a darker color are correlated with more pesticides, and arrangement around the circle is ordered from those with the most correlations (dicofol, darkest color) to the least (petroleum hydrocarbons, lightest color). Dicofol, for example, resulted in significant mDA cell death in the iPSC-model and is therefore shown as a teal highlighted diamond. It was also both (1) the most statistically significant mDA toxic pesticide in the PD PWAS (FDR = 4.2e-05) and therefore shown as the largest diamond, and (2) correlated above R > 0.45 with the most other PWAS-implicated pesticides (n = 24 pesticides), and therefore shown as the darkest color. Note, pesticides that did not correlate across layer at R > 0.45 are not shown on the wheel. Diquat dibromide, for example, was mDA toxic, however, the strongest correlation diquat displays with another pesticides was R = 0.14. Source data are provided as a Source Data file.

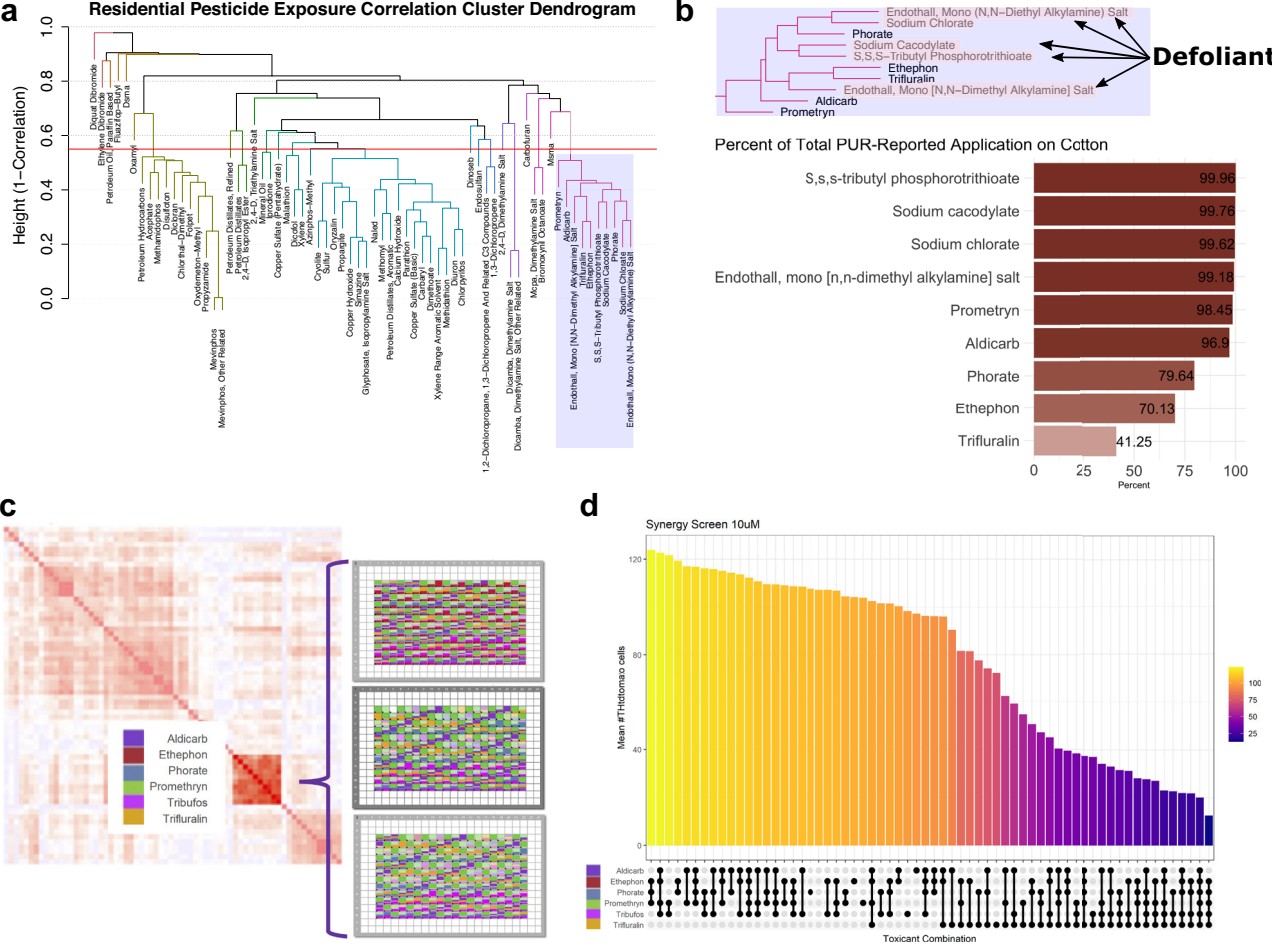

**Fig. 6 | Co-exposure of pesticides in the Cotton Cluster demonstrates evidence of synergistic toxicity. a** Cluster dendrogram from hierarchical clustering of the PWAS-pesticides using residential address-based exposures (clusters cut at height = 0.55), to identify groups of highly correlated pesticides for co-exposure analysis. **b** Cotton cluster: cluster identified as of interest, as it includes two of the three most significantly PD-associated pesticides in the PWAS based on FDR (sodium chlorate and prometryn) and half of the pesticides in the cluster have a use type of defoliant. The bar graph shows the proportion of all agricultural application records in the tri-county study area for each pesticide used on cotton, with the darker color representing larger proportions. For example, 99.96% of the reported S,S,S-tributyl phosphorotrithioate (aka tribufos) applications were on cotton.

**c** Schematic outlining how pesticides from a single co-exposure cluster (cotton cluster) were recombined in all possible combinations of six pesticides using an HP Digital dispenser on sorted dopaminergic neurons plated into a 384 well format, similar to the survival assay described in Fig. 4. **d** An upset plot was used to sort and display the most toxic and least toxic combinations. Y-axis shows number of THtdTomato+ neurons at day 11 following treatment. Ball and stick connections along the X-axis indicate co-treatments with a ball indicating treatment with a given pesticide. Cooler (purple) colors represent lower relative cell counts while warmer (yellow) colors represent higher cell counts. DMSO control condition is depicted by the x-value lacking any ball and stick marker. N = 4 biological replicates. Source data are provided as a Source Data file.

commonly applied to the same fields (i.e., from the same exposure cluster). To test effects of co-exposure to the cotton cluster, we performed a survival assay on sorted mDA neurons from the same PD patient derived line described above using all combinations for a subset of pesticides in this cluster. The schematic in Fig. 6c illustrates how pesticides from a single co-exposure cluster (shown in more detail in Fig. 5a) were combined in all possible combinations in an exposure matrix. We used a live imaging survival assay and treatment paradigm similar to that used for data presented in Fig. 4. A digital dispenser (Hewlett Packard) was used to dispense pesticides.

A 10 μM dose was chosen for this assay as an intermediate dose, permitting detection of additive or synergistic effects. In contrast, combinations of pesticides at 30 μM proved excessively toxic during assay development. Pesticide combinations were arranged in descending order by number of surviving neurons after treatment (Fig. 6d). The average of four independent biological replicates is shown. *P*-values for individual pairwise comparison with adjustment for multiple testing are provided in Supplementary Data 14. Combinations involving trifluralin caused more mDA neuron cell death (gold bar) and there was potential synergy with tribufos (S,S,S-tributyl phosphorotrithioate) as this combination resulted in the most mDA neuron cell death of all the pairwise combinations. Trifluralin alone at 10 μM produced a 32% decrease in mDA neurons compared to DMSO, while tribufos produced an 8% decrease (neither statistically significant in this assay), but in combination they produced a 65% decrease compared to DMSO and were significantly different from the individual treatments at $p = 0.003$ for the comparison to tribufos alone and $p = 0.048$ for the comparison to trifluralin alone. Performance in this assay showed no clear correlation to the odds ratio for PD risk or FDR cutoff level described in Fig. 2b.

## Trifluralin reduces spare capacity of mitochondria in mixed PD-patient-derived dopaminergic neuron cultures

Identification of trifluralin in the PWAS screen, in vitro toxicity assay, and demonstration of synergistic action prompted a deeper investigation into potential mechanisms by which this pesticide causes mDA neuron cell death. Trifluralin was previously shown to increase α-synuclein fibril formation in a cell-free system[31]. Additional biochemical and metabolic assays were utilized to examine the effects of trifluralin on mDA neurons. These assays necessitated alterations in assay timing and conditions from the original live imaging survival assay. Neurons used for biochemical assays could be matured longer in vitro prior to exposure and harvesting because the clumping that occurs frequently with prolonged two-dimensional culture does not impact biochemical or metabolic assays in the way it complicates image analysis. PD patient-derived mDA enriched cultures were treated with sublethal doses of trifluralin (6 μM) for two weeks to assess for accumulation and expression of phosphorylated α-synuclein (pS129). The timing is similar to the duration assessed in our recently described iPSC-based inclusionopathy models[32]. Western blot revealed very low levels of pS129 expression in both control and trifluralin treated conditions and no significant difference in pS129 was detected (Supplementary Fig. 13, experiment performed three times, representative blot shown). In the absence of a robust α-synuclein-related phenotype, we next examined mitochondria. While trifluralin is annotated as a mitosis inhibitor in targeted grasses and weeds, mDA neurons are known to be exquisitely sensitive to mitochondrial dysfunction, including by toxicants that target the mitochondria directly[33,34]. We thus investigated whether trifluralin could also result in mitochondrial dysfunction in mDA neurons. mDA neuron-enriched cultures were assessed for effects on mitochondrial function at 55 days in vitro at a time when more mature, arborized neurons are observed. We quantified relative mitochondrial subunit abundance, mitochondrial respiration, and function of the oxidative phosphorylation pathway in mDA neuron cultures. We measured mitochondrial subunit abundance

after 24 h exposure to 30 μM trifluralin in 3 biological replicates and demonstrated a 50% reduction of Complex I (NDUFB8) and Complex IV (COX II) when compared to DMSO treated controls (Fig. 7a). There was no significant reduction or increase of the expression of other complexes in the mitochondrial respiratory chain (see original blot in Source Data file Fig. 7).

For assessment of mitochondrial function, we used the Seahorse assay, which involves sequential addition of mitochondrial complex inhibitors (Oligomycin [complex V], FCCP – uncoupler – and Rotenone [complex I]/Antimycin-A [complex III]) and measurement of media acidification and oxygen consumption rate (OCR). The effects of trifluralin were assessed in a dose-response manner in three biological replicates (Fig. 7d). We calculated mitochondrial respiration parameters (basal respiration, ATP production, maximal respiration and spare capacity) and found, with the exception of ATP production, all parameters were decreased with exposure to trifluralin at the concentrations used in the assay. The respiratory capacity of differentiated neurons was reduced to 9% when cells were treated with trifluralin at 90 μM, 26% at 60 μM, and 72% at 30 μM trifluralin compared to DMSO control, with the highest two doses being significantly different from DMSO control (Fig. 7e). The effect of trifluralin on cellular respiration was dose and time-dependent (Fig. 7d, e). Mitochondrial function was tested at early time points (6 h after treatment) with a goal of targeting more direct metabolic effects from the pesticide treatment. These effects are expected to occur at shorter time points compared to overt toxicity. We also wanted to avoid prolonged exposure that could lead to nonspecific mitochondrial damage or dysfunction to accumulate. Mitochondrial protein content, on the other hand, was assayed to measure steady state changes that could result from exposure, and thus assayed at a later time point. We followed literature precedent for these assays[35].

We also exposed mDA neurons to ziram as a control. Ziram has been implicated in PD previously and its toxicity has been tied to an inhibitory effect on the E1 ligase in mDA cultures without any described mitochondrial phenotypes[28]. Ziram exposure for 2 or 6 h at a dose greater than the LD50 from survival assays (300 nM) did not significantly alter the oxygen consumption rate (Fig. 7f) or the mitochondrial reserve capacity (14% increase in 6 h exposure compared to DMSO, Fig. 7g). This is in contrast to the effect of trifluralin at a similar ratio to its LD50 (60 μM). These data implicate mitochondrial dysfunction in trifluralin-mediated mDA neuron cell death, reinforcing that pesticides toxic to mDA neurons exhibit distinct mechanisms of toxicity.

## Discussion

Pesticides have been definitively linked to Parkinson's disease etiology by prior studies[36]. However, most specific agents used in agriculture have not been assessed for potential influence on PD. Therefore, we established a field-to-bench paradigm which combined two distinct approaches: 1) broad epidemiologic screening of hundreds of pesticides for association with PD; and 2) in vitro evaluation of associated pesticides in mDA neurons. This approach permitted testing of epidemiologic hits for direct effects on dopaminergic neurons to better identify and classify PD-relevant pesticides.

The record-based exposure assessment and epidemiologic screening implicated 68 pesticides with PD. To identify potential direct effects on mDA neurons, we coupled our PWAS screen to systematic analysis of the hits in human mDA neurons derived from iPSCs of a patient with triplication of the wild type α-synuclein locus (SNCA). These cells capture human biology that may differ in critical ways from rodent or other human cell lines, most dramatically with the biology of dopamine oxidation[6].

We tested 39 of the epidemiologically implicated pesticides in vitro. Ten resulted in substantial mDA neuron cell death at 30 μM. The identified toxicity in these neurons represents a unique and

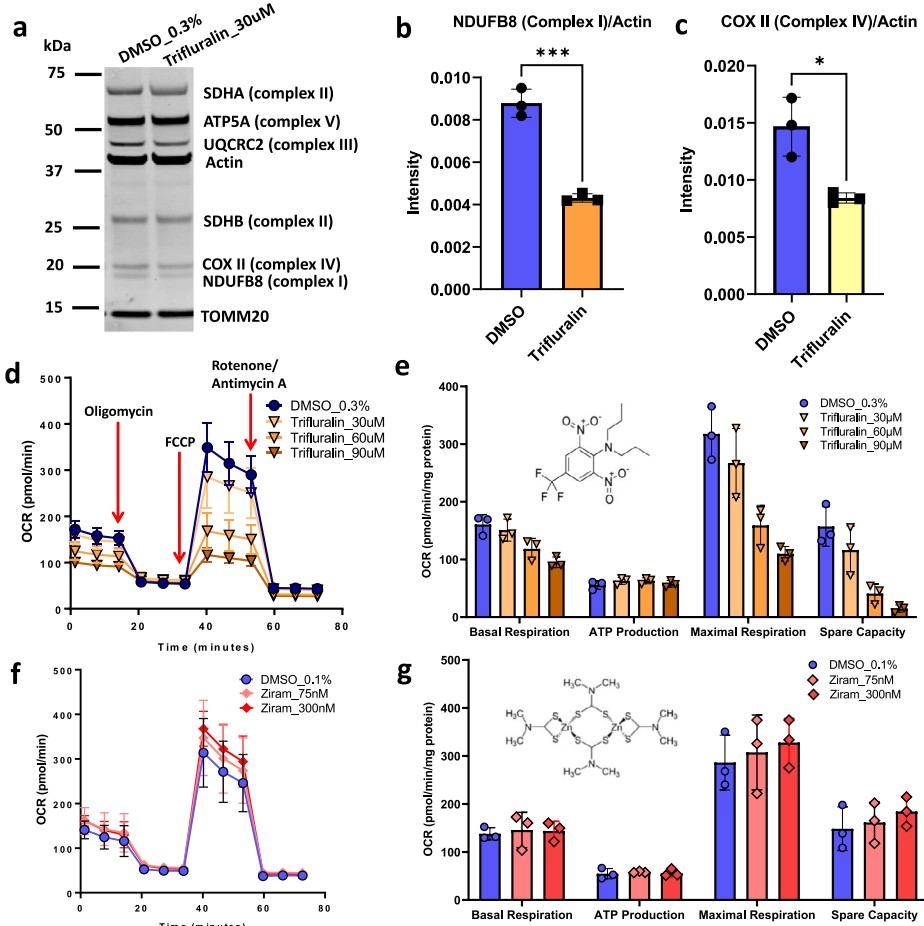

**Fig. 7 | Trifluralin alters mitochondrial subunit abundance and oxygen consumption rate.** Effect of Trifluralin in mitochondria subunit and effect of Trifluralin and Ziram in Mito-stress assays. **a** Western blot analysis of respiratory chain complexes for differentiated SNCA-triplication neurons at DIV 65, exposed to 0.3% DMSO and 30 μM Trifluralin for 24 h. Uncropped blots in Source Data. **b** Quantification of Complex I and **c** Complex IV from blot in **a** normalized to actin, (●) 0.3% DMSO and (■) 30 μM Trifluralin. T-test for mitochondria subunit, $p < 0.001$ (***), $p < 0.01$ (*), $n = 3$ biologically independent replicates, **d** Measurement of Oxygen Consumption Rate curves on Mito-stress assay for the dose-response effect of (●) DMSO 0.3% and (▼) Trifluralin (30 μM, 60 μM and 90 μM) on SNCA-triplication differentiated neurons at DIV 65 and after 6hrs exposure. **e** Metabolic parameters (Basal respiration, ATP production, maximal respiration and spare respiratory capacity) calculated from **d**. 2-way ANOVA for Mito-stress assays, $p < 0.0001$ (****); Dunnett's multiple comparisons test $p < 0.0001$ for DMSO vs. 60 μM and DMSO vs. 90 μM Trifluralin. DMSO vs. 90 μM Trifluralin not significant, $p = 0.099$. **f** Oxygen Consumption Rate curves on Mito-stress assay (●) 0.1 % DMSO and (♦) 75 nM and 300 nM Ziram exposure on SNCA-triplication differentiated neurons for 6 h. **g** Metabolic parameters for the conditions described in **f**. There were no significant differences among treatment conditions with Ziram ($p = 0.4326$). Errors bars represent standard deviation, $n = 3$ biologically independent replicates for 7d–7g. Source data are provided as a Source Data file.

previously unappreciated common characteristic of these pesticides – they do not have another previously described shared characteristic or property. Interestingly, our experiments identified propargite and confirmed previous work with this pesticide in mDA neurons from a different iPSC line[37]. Other pesticides highlighted in our studies, including diquat dibromide and the copper sulfates, also have persuasive biologic links to PD[38–43]. Notably, however, we implicated several pesticides as toxic to mDA neurons that have not previously been linked to PD or neuronal toxicity, including folpet, naled, endosulfan, and endothall. Furthermore, 8 of the 10 mDA toxic pesticides are still registered for use with the US EPA. Five of the pesticides also showed selectivity for neurons given that they did not produce substantial toxicity in cardiomyocytes. These will be the subject of future testing on other CNS cell types, co-culture systems, and 3D organoids. Given both the public health relevance due to current use and findings presented here, these agents certainly warrant additional investigation.

In the current manuscript, we focused on the differential effects of pesticides at the screening dose on mDA neurons as a first layer of specificity. We then confirmed this result in an additional cell line to assess for generalizability across dopamine neurons from different sources. Finally, we assessed cardiomyocytes to identify which pesticides are likely to be toxic to any cell rather than specific to neurons. It will be important in future investigations to develop more insight into what constitutes clinically relevant dose exposures. This is a nontrivial issue – modeling a chronic exposure "in the dish" is challenging. Developing more sensitive assays that can assess for phenotypes at low micromolar or nanomolar concentrations will be important. Functional assays, for example of neuronal activity, may be more sensitive than looking at overt toxicity. A useful indicator of a relevant toxicant may be one that is toxic to a mutant mDA neuron but not an isogenic corrected one. Finally, where possible it will be important to better model toxicant exposure in vivo in chronic exposure paradigms in animal models. Postmortem brain examination, where possible, may provide important clues to the steady state concentrations of toxicants and their metabolites in human brain as a benchmark for modeling these exposures in the laboratory setting.

Beyond individual agents, an important use of our platform is to screen pesticide co-exposures and understand synergistic

interactions. Analysis of co-treatment with components of the cotton cluster indicated that co-exposures involving trifluralin produced substantially more mDA cell death than any of the components alone. Trifluralin has been previously linked to PD in the Agricultural Health study, making this a compelling pesticide for more in depth mechanistic investigation[44].

Using functional assays in mDA neurons, we implicated mitochondrial dysfunction in trifluralin-mediated toxicity. Trifluralin is part of the dinitroaniline family of herbicides, known to cause disruption of cell division in plant cells and protozoa via de-polymerization of microtubules[45], binding to tubulin in higher plants and protozoa[46–48], but not mammalian or human tubulin[49]. We are not aware of any reports directly linking trifluralin to mitochondrial dysfunction in neurons, suggesting that the use of human mDA neurons as a model can uncover previously unrecognized toxicity pathways for pesticides. Our results support an effect of trifluralin on neuronal respiration and mitochondrial function, consistent with an extensive literature documenting mitochondrial dysfunction as fundamental to PD[50–53]. Imaging-based assays of mitochondrial potential or neuronal activity will enable assessing toxic effects of much lower doses of trifluralin. It will be interesting to understand whether the effects of pesticides are α-synuclein-dependent. Additional work is required to understand why a pesticide like trifluralin causes α-synuclein fibril formation in a cell-free system but does not cause increased phosphorylated α-synuclein[31]. This can be assessed in more tractable cell lines, with isogenic mutation-corrected, and α-synuclein knockout lines[32]. For some pesticides, like paraquat, a toxic effect through nitrosative stress has already been tied directly to α-synuclein proteotoxicity and biology[8,20]. A similar approach will be valuable to undertake for the top-hit pesticides identified in this study.

Several considerations for interpreting this wide-scale pesticide screening are noteworthy. First, we view the population-based screening as a first step to prioritize agents for more in-depth research. It does not necessarily suggest a causal role for PD in all those that screened positive, nor does it absolve those which failed to screen positive for a role in PD. As case in point, pesticide exposures were often highly correlated in the epidemiologic study. For instance, with the PWAS-implicated pesticides, we observed that even if the pesticide itself did not result in significant mDA cell death in the iPSC-model, its exposure levels correlated relatively highly with a pesticide that did. This presents some interesting interpretations for the lack of significant mDA neuron death in vitro for many of the pesticides implicated by the PWAS. First, such pesticides may in fact have no direct influence on PD and the association was simply driven by correlation with toxic co-exposures. However, the pesticides may also influence PD through other disease-relevant pathways besides mDA toxicity or be involved in key mechanisms that were not recapitulated in our cell model. Finally, the pesticides could be toxic but only in combination with other pesticides, as the cotton cluster analysis suggested.

With regard to the epidemiologic exposure assessment, while the GIS-based method is uniquely informative and has been previously validated[54], it still likely allows for some level of non-differential exposure misclassification (see Supplementary Materials). Conversely, the assessment did not rely on self-report, an advantage that cannot be overstated in population-based research. Moreover, coupling the epidemiologic screen with experimental study provides confidence in both the field and bench aspects of the paradigm.

With the in vitro modeling, limitations include relative immaturity of cultured dopaminergic neurons; a lack of a blood-brain barrier (BBB) and a greatly simplified cellular environment lacking astrocytes, microglia, endothelial cells, and circulating factors. More advanced models, such as triculture systems that reconstruct the neuroimmune axis and "BBB on a chip" methods will permit investigation of more complex inter-cellular interactions and render the platform more

physiologic[55,56]. Multiplexed iPSC co-culture will even enable the investigation of whole populations of patients in a single dish[57,58].

As noted above, dosing is an important consideration in our study. The pesticide doses used in our study, while comparable to other published reports, are high[8,26,27]. We used high doses to accelerate the onset of pathologies in the dish by using higher concentrations for shorter times rather than years worth of lower-level exposure. Our in vitro models could also underestimate the toxicity of pesticides that require metabolism in the liver or by glial cells to generate harmful toxic metabolites. Additionally, with our live imaging system, we have chosen a straightforward and dramatic phenotype as the readout for pesticide toxicity–cell death. More subtle phenotypes, such as α-synuclein phosphorylation, abnormal neuronal activity, and dopamine oxidation will be important to detect effects of low doses of pesticides.

The California Pesticide Use Report system has enabled us to begin to understand the breadth of pesticide application in agricultural regions and investigate how it relates to PD in the community. Here, establishing a field-to-bench paradigm, we have combined record-based exposure assessment with a tractable and screenable iPSC-derived dopaminergic neuron model to identify and classify PD-relevant pesticides.

Our comprehensive, pesticide-wide association study has implicated both a variety of individual pesticides in PD risk and suggested relevant co-exposure profiles. Coupling this with direct testing in vitro on dopaminergic neurons, we have pinpointed pesticides that were directly toxic to human dopaminergic neurons. Further, real-world co-exposure data has allowed us to develop co-exposure paradigms "in the dish" and establish which combinations of pesticides can indeed lead to greater, synergistic mDA toxicity. Ultimately, we have identified pesticides that are both ostensibly mDA-toxic and pesticides that are not mDA-toxic, but, nonetheless, associated with increased risk of PD. In time, this field-to-bench approach will enable us to further tie cellular pathologies back to this epidemiologic and environmental data, to mechanistically understand the individual and combinatorial effects of pesticides and their interactions with genetic susceptibility. Collectively, these studies will inform the judicious use of pesticides in agriculture.

## Methods

### Compliance

All data presented and experiments described herein are conducted in accordance with the Institutional Review Boards of Brigham and Women's Hospital (IRB#2019P002015), Harvard University Faculty of Arts and Sciences (IRB#19-0736), and University of California, Los Angeles (IRB#21-000256 and IRB#11-001530). Informed consent was obtained from all epidemiologic study participants.

### PEG study population

To assess pesticide and PD associations in the PWAS, we used the Parkinson's Environment and Genes (PEG) study ($n = 829$ PD patients; $n = 824$ controls). PEG is a population-based PD case-control study conducted in three agricultural counties in Central California (Kern, Fresno, and Tulare)[16]. Participants were recruited and enrolled in two waves: wave 1 (PEG1): 2000–2007, $n = 357$ PD patients, $n = 400$ population-based controls; wave 2 (PEG2): 2009–2015, $n = 472$ PD patients, $n = 424$ population-based controls. Patients were early in their disease course at enrollment (mean PD duration at baseline 3.0 years (SD = 2.6)), and all were seen by UCLA movement disorder specialists for in-person neurologic exams, many on multiple occasions, and confirmed as having idiopathic PD based on clinical characteristics[17]. Population-based controls for both study waves were required to be >35 years of age, have lived within one of the three counties for at least 5 years before enrollment, and not have a diagnosis of PD. More information about enrollment can be found in the Supplementary Materials. Characteristics of the PEG study subjects

are shown in Supplementary Data 15. The PD patients were on average slightly older than the controls and had a higher proportion of men, European ancestry, and never smokers.

## PEG pesticide exposure assessment

We estimated ambient exposure to specific pesticide active ingredients (AIs) due to living or working near agricultural pesticide application, using record-based pesticide application data and a geographic information systems-based model[15]. We briefly describe our method but provide more detail in the Supplementary Materials.

Since 1972, California law mandates the recording of all commercial agricultural pesticide use by pest control operators and all restricted pesticide use by anyone until 1989, and then (1990-current) all commercial agricultural pesticide use by anyone, to the PUR database of the California Department of Pesticide Regulation (CA-DPR). This database records the location of applications, which can be linked to the Public Land Survey System, poundage, type of crop, and acreage a pesticide has been applied on, as well as the method and date of application. We combined this database with maps of land-use and crop cover, providing a digital representation of historic land-use, to determine the pesticide applications at specific agricultural sites[59]. PEG participants provided lifetime residential and workplace address information, which we geocoded in a multi-step process[60]. For each pesticide in the PUR and each participant, we determined the pounds of pesticide applied per acre within a 500 m buffer of each residential and workplace address each year since 1974, weighing the total poundage by the proportion of acreage treated (lbs/acre).

We were interested in long-term ambient exposures, and thus, considered the study exposure window as 1974 to 10 years prior to index date (PD diagnosis for cases or interview for controls) to account for a prodromal PD period. The exposure windows, which were on average 22 years for residential exposure and 18 years for workplace exposure, covered a very similar length and temporal period on average for patients and controls of each wave (study window comparisons shown in Supplementary Data 1 and 15). To assess exposure across the study window of interest for each pesticide, we averaged the annual lbs/acre estimates in the study window (e.g., for a participant with 22 years of exposure history, lbs/acre estimates for all 22 years were averaged), only using years for which the participants provided address information. This approach created one summary estimate of the average pounds of pesticide applied per acre per year within the 500 m buffer for each pesticide, which was estimated at residential and workplace locations separately for each participant. We log transformed the estimates offset by one, centered, and scaled the estimates to their standard deviations (SD).

## Pesticide regulation and toxicity classification

We linked each of the 288 pesticides included in the PWAS to chemical, regulatory, and toxicity information using publicly available databases. We used the Pesticide Action Network (PAN) database to determine the chemical class (e.g. organophosphorus) and use type (e.g. insecticide) for each pesticide[19]. We linked each pesticide associated with PD to registration status in the United States and European Union to highlight current, active agricultural use versus historical, past use. We obtained information on pesticide regulation using the U.S. EPA pesticide label database, U.S. EPA cancellation reports, California Product Label Database, and the European Commission database, Rotterdam Notifications, and PAN. We further interrogated the PAN database to link information on known toxicity[19]. This includes whether the pesticide is drift prone based on vapor pressure, a groundwater contaminant, acutely toxic, a cholinesterase inhibitor, an endocrine disruptor, a carcinogen, or a developmental or reproductive toxicant. This information is based on public databases from government and international agencies, such as the EPA, California Prop 65 lists, World Health Organization, and International Agency for Research on Cancer.

To identify a "most toxic" set of pesticides, PAN North America has designated certain pesticides as 'bad actors', if the pesticide meets any of the following criteria: known or probable carcinogen, reproductive or developmental toxicant, neurotoxic cholinesterase inhibitor, known groundwater contaminant, or a pesticide with high acute toxicity. Detailed methods on the toxicity and PAN designations can be found on the PAN website (https://www.pesticideinfo.org/).

## iPSC reporter generation

Method for derivation of THtdTomato knockin clones was adapted from Ahfeldt et al. [14]. iPSC lines derived from a male patient with triplication at the α-synuclein locus (referred to as "SNCA triplication") was used for this[20,21]. A second iPSC line, generated from a distinct male donor unrelated to the SNCA triplication donor, contains two copies of wild-type α-synuclein was modified according to the same protocol summarized below. The line originally harbored an α-synuclein point mutation (E46K). We introduced the TH reporter and then genetically corrected it to create a two-copy SNCA wild-type line. THtdTomato insertion is initiated with nucleofection of two plasmids containing sequence coding for gRNAs and a third targeting plasmid were co-nucleofected using the Amaxa 4D nucleofection system on a Lonza nucleofector. Targeting plasmid contained: the targeting vector with a TH homology arm followed by tdTomato, a 2 A self-cleaving peptide sequence, a WPRE sequence, floxed puromycin selection cassette, and TH homology arm. Cells were allowed to recover for one day prior to initiation of puromycin selection. Surviving colonies were expanded and pooled for nucleofection of a pCAG-CRE:GFP vector allowing for Cre:LoxP based excision of the puromycin selection cassette and isolation by FACS of GFP + cells. Cells were plated at clonal density and resulting colonies were expanded out and subcloned. A 5' genotyping PCR producing a 626 bp product confirmed proper insertion of the 5 prime end of the reporter construct (5PrimeF: ACA TCC CCT GCT TGT TTC AA; 5PrimeR: AGC CCT CTA GCC TCA TCC TC) and a 3' genotyping PCR producing an 878 bp product (3PrimeF: TCC CTC AGA CCC TTT TAG TCA; 3PrimeR: GAG CCT CTG GAG CTG CTT G) confirmed proper insertion of the 3 prime end of the reporter into exon 14 of the TH gene. PCR products were Sanger sequenced and aligned to the sequence expected after successful targeting. A third PCR designed to produce a 285 bp product was performed to evaluate the unmodified TH allele and assess for NHEJ errors (NHEJF: CGT GAA GTT CGA CCC GTA CA; NHEJR: ACA GCT GTT GCG CTG AGA AG). Colonies passing the above quality control were differentiated into dopaminergic neurons[12,14,61] and assessed for proper expression of THtdTomato insert with both live imaging and post-fixation immunoco-localization of tdTomato reporter (rabbit anti-RFP, Rockland, 1:500) with tyrosine hydroxylase (sheep anti-TH, Pel-Freeze #960101, 1:500). Neurons were fixed with 4% paraformaldehyde for 20 min at room temperature after 35 days of differentiation as embryoid bodies and 13 days in adherent culture. Clones that pass PCR, sequencing, and differentiation quality control steps are karyotyped to assess for any abnormalities.

## Dopaminergic differentiation

Dopamine neurons were generated from the THtdTomato modified iPSCs in accordance with published protocols, using minimal modification[12,14,61]. iPSC cultures were maintained in growth media (StemFlex, mTESR plus, or mTESR media). On day 0 of differentiation, confluent iPSC cultures on matrigel or geltrex were dissociated into single cells using Accutase incubated for 5–7 min at 37 degrees following a 0.5 mM EDTA in PBS wash. Accutase reaction was quenched with growth media and single cells were centrifuged, resuspended in growth media supplemented with Y-27632 at 10 μM and FGF-2 at 20 nM. Cells were plated at a density of $1 \times 10^6$ cells/mL in 15 cm uncoated plates (30 mL per plate) to form embryoid bodies. An additional 30 mL of growth media is added the following day. On day 2, EBs are collected into 50 mL conicals and spun at 200 x g for

3 min. Media is aspirated and EBs are replated in differentiation media (DMEM:F12 media mixed 1:1 with Neurobasal media, with PenStrep, B27 supplement without vitamin A, N2 supplement, 2-mercaptoethanol and glutamax) supplemented with 100 nM LDN-193189 and 10 μM SB431542. On day 3, 30 mL of differentiation media is added with SAg (1 μM) in addition to LDN-193189 and SB431542. Differentiation media is changed daily until day 5 at which point CHIR99021 is added (3 μM) and media changes are then performed every other day. SB431542 and SAg are withdrawn on day 9. LDN-193189 is withdrawn on day 13 and BDNF (20 ng/mL), GDNF (20 ng/mL), TGFbeta3 (25 ng/mL), dibutryl cAMP (0.5 mM), and DAPT (10 μM) are added. CHIR99021 is withdrawn on day 15 and TGFbeta3 is withdrawn on day 17. Spheres are maintained on uncoated plates with three times weekly media changes until dissociated for FACS and biochemical assays on day 35–42.

### Dissociation, FACS, and live cell toxicant survival assay

Spheres are collected in a 15 mL conical tube from suspension culture plate, washed with PBS and resuspended in 2 mL of 0.25% Trypsin EDTA with 25 ng/mL of DNAse added prior to incubation at 37 degrees in water bath or rotating shaker for 5–7 min. 500 uL of FBS is then added to stop the reaction. Following a PBS wash, the EBs are triturated 5–7 times with a P1000 in a trituration solution (PBS with 5% FBS, 25 mm Glucose, 1x glutamax). Cells are then washed with PBS and pelleted at 300 x g for 5 min. Washes are repeated 3–5 times prior to plating or FACS sorting. Large clumps and aggregates are filtered using a 35 μM CellTrics filter. Y-27632 at 10 μM is present for sorting and collection in differentiation media. A MoFlo Astrios and MoFlo XDP (Beckman-Coulter, both equipped with 100um nozzle at 30 psi, running Summit Software V6.1.16945; analysis with FloJo 10.6.1) were used to sort single, live THtdTomato + neurons based on scatter profile, pulse width, exclusion of Sytox Red dye, and tdTomato fluorescence. The brightest 30–40% of cells are included in order to minimize non-neuronal cell types or immature/neuronal progenitors expressing a low level of the THtdTomato reporter. Sorted cells are plated onto polyornithine, poly-D-lysine, laminin, and fibronectin coated assay plates (Greiner 384 well plates) for survival assays. Sorted cells were plated at a density of $4 \times 10^3$ cells per well of a 384 well plate in a total volume of 45 uL on the day of plating with Y-27632. Media wash and transition to Fluorobrite (GIBCO) live imaging media was performed the following day using an Apricot Personal Pipettor (Apricot Designs) leaving 90 uL of fresh media per well. The first pesticide/toxicant treatment was performed two days after plating (Supplementary Figure 6). At five days after plating, half the media is aspirated, 45 uL of fresh media are added and treatment is repeated. At nine days after plating, half the media is aspirated, 45 uL of fresh media are added but treatment is not repeated. Live imaging is performed with a live cell chamber-equipped to maintain 5% CO2 and 37 degrees on an IXM High throughput microscope (Molecular Devices, MetaXpress imaging software version 6.5.3.427) using a x10 objective, a Texas Red filter (excitation 560/32 nm; emission 624/40) and imaging four fields per well, resulting 72% coverage per well. Images were acquired immediately prior to treatment, at 7 days after first treatment and at 11 days after first treatment.

### Image analysis

Images were imported into Columbus (Perkin Elmer, version 2.9.1) analysis software. Differential brightness and roundness criteria permit the use of nuclei detection algorithms to detect the endogenous fluorescent reporter for accurate selection of cell soma with exclusion of neurites. The *Find Nuclei* script was used with method M, diameter was set at 13 μM with splitting sensitivity of 0.15 and common threshold of 0.2. The nuclei detection algorithm was further refined to eliminate debris and doublets by selecting nuclei with area >40 μM and <400 μM, and roundness >0.7. Additional brightness criteria (pixel intensity >500) were used to generate reproducible neuron cell body detection scripts that exclude debris and dead cells. Image analysis was performed in two steps to improve assay to assay reproducibility. The first step determined the average pixel intensity for detected cells that met size and roundness criteria. Average and standard deviation of fluorescence intensity was then determined for the control wells treated with DMSO. A second analysis was then designed to count all cells brighter than three standard deviations above the average fluorescence intensity in the control wells and measure neurites. These objects were counted as THtdTomato positive cells for analysis. Built in neurite detection algorithms were applied in order to identify neurites based on THtdTomato signal in these cellular processes using the Find Neurites, CSIRO neurite analysis method. For this neurite detection method, the following settings were used: smoothing window 3px, linear window 15px, contrast >1, diameter >/= 7px, gap closure distance </= 5px, gap closure quality 0, debarb length </= 15px, body thickening 5px and tree length </= 20px. Neurite analysis generated a sum of total neurite length per field analyzed which was used as the primary metric for neurite analysis.

### Toxicant library generation

All compounds were ordered from Sigma Aldrich as the PESTANAL analytical standard whenever possible with the following exceptions: Tribufos (S,S,S-tributyl phosphorotrithioate) obtained from Fisher/Crescent Chemical; MSMA (Monosodium acid methane arsonate sesquihydrate) obtained from Santa Cruz biotechnology: oxydemeton-methyl obtained from Santa Cruz biotechnology. Based on available solubility data, compounds were dissolved in DMSO, water, or ethanol to a working dilution of 30 mM. A subset had poor solubility that required a more dilute stock solution (15 mM or less). A limited set of compounds identified in the PWAS analysis were omitted due to high dermal/inhalation toxicity in mammals, inability to obtain a highly pure formulation, or inadequate solubility in water, DMSO, or ethanol. Working stocks of compounds were pipetted into multiwell template plates and serial dilution was performed with an Apricot Personal Pipettor. Ethanol solubilized compounds were diluted 1:1 with DMSO to improve accuracy of pipetting and permit a parallel workflow to the DMSO and water plates. Individual compound plates were then generated from template plates following serial dilution and stored at −20 °C until day of treatment. Apricot personal pipettor equipped with 125 uL volume disposable tips was used to perform dilution of compounds in culture media and treatment of dopaminergic neurons.

### Viability assays

mDA neurons were sorted and pesticide treatments were performed as described. At eleven days after the first toxicant treatment, media was removed and the CellTitreGlo buffer was added directly to the assay plate using an Apricot Personal Pipettor. CellTitreGlo assay was scaled down from the manufacturer's recommended volumes to accommodate a 384 well format. Luminescence was measured on a Molecular Devices Spectramax plate reader (using SoftMax 5.4 pro software) 15 min after buffer addition.

### Cardiomyocyte differentiation

The obtained hiPSC clones were cultured in StemFlex cell culture media (ThermoFisher Scientific, A3349401) in 6 well plates coated with Geltrex LDEV-Free Reduced Growth Factor Basement Membrane Matrix (ThermoFisher Scientific, A1413202). Once confluent, the hiPSC cells were differentiated into human CMs utilizing a chemically-defined cardiomyocyte differentiation protocol[62]. Briefly, hiPSCs were treated with 6 uM CHIR99021 (Tocris, 4423) for 3 days in RPMI 1640 (Thermo Fisher Scientific, 11875119) with B27-insulin (Thermo Fisher Scientific, A1895601). Then cells were treated with the 2uM Wnt inhibitor C59 (Tocris, 5148) in RPMI/B27- for another 2 days. Between 5–10 days of differentiation, RPMI/B27- media was used and changed

every other day, on day 11 cells were switched to RPMI/B27 + insulin and beating was observed. To improve the CM purity, cells were cultured in RPMI/B27 + insulin without glucose, for 3 days. After purification, the iPSC-CMs were dissociated with TrypLE 10x (Thermo Fisher Scientific, A1217703) and replated in 6-well Matrigel coated plates at a density of $3 \times 10^6$ per well in RPMI/B27 + containing 10% KOSR and ROCK inhibitor Y-27632 (Tocris, 1253). After 2 days, cells were cultured again in RPMI/B27 + without glucose for 3 days prior to switch to 3 ml of RPMI/B27 + insulin. Cells where cultured for an additional week with media changes every 2–3 days prior to subsequent imaging studies.

## Cardiomyocyte cell count assays

hiPSC-CMs were plated on Geltrex at 40,000 cells per well of a 384-well plate (Greiner Bio-One) in 75 μl of RPMI/B27 + insulin containing 10% KOSR and ROCK inhibitor Y-27632. The next day media was changed to RPMI/B27 + insulin final volume 90 μl and cells were maintained for a minimum of 5 days prior to drug treatment and imaging. For the analysis, 45 μl of media was removed and hiPSC-CMs were loaded with 45 μl 2x TMRM (T668, ThermoFisher Scientific) with Hoechst 33258 (ThermoFisher Scientific). After 10 min incubation in a 37 °C 5% CO2 incubator, cells were washed by removing 90 μL of media and replacing it with 90 μL RPMI/B27 + insulin. Hoechst 33258 signal was acquired across 4 tiled fields per well at 20x, using an EVOS M7000 (ThermoFisher Scientific). Nuclei were counted using onboard software.

## Western blot

For western blots, protocol was performed as described with minimal modifications[32]. Briefly, 20–30 μg of protein per sample was subjected to SDS-PAGE using NuPAGE 4–12% Bis-Tris protein gels in NuPAGE MES SDS Running Buffer, electrophoresed at 150 V for 1 h. Dry transfer from polyacrylamide gel to PVDF membrane was conducted with the iBlot 2 Gel Transfer Device (Thermo Fisher) using preset P0 program (20 V 1 min; 23 V 4 min; 25 V 2 min). The membrane was fixed in 4% paraformaldehyde in PBS for 30 min at room temperature with orbital shaking and washed three times for 5 min with PBS. Membranes were blocked in 5% nonfat milk in PBS 1 h with orbital shaking and subsequently incubated overnight in 5% block with 0.1% Tween20 and the respective primary antibodies at the desired dilution (see below for antibody information and dilutions), at 4 degrees C with orbital shaking. After four washes for 5 min in 0.05% Tween20/PBS, membranes were incubated with the HRP-conjugated secondary antibody at 1:10,000 in blocking solution at room temperature with orbital shaking. After four washes for 5 min in 0.05% Tween20/PBS, the blot was incubated for 5 min in chemiluminescence detection solution (Bio-Rad Clarity Western ECL Cat# 1705060). Signal detection was performed on an iBright imaging system (Invitrogen). Primary antibodies utilized include rabbit anti-alpha synuclein phospho S129 (Abcam Cat # ab51253; 1:1000), mouse anti-Alpha Synuclein (Clone: 42, BD Transduction Laboratories Cat. # 610787, 1:1,000), and mouse anti-GAPDH (Clone 6C5, Millipore Cat# MAB 374; 1:15,000).

## Combinatorial treatments

The combinatorial treatment plate maps were designed and performed using a D300e Digital Dispenser (Hewlett Packard) equipped with T8 and D4 dispenseheads. HP software was used to generate plate maps using the "synergy" function for 384 well format that produced all possible combinations of 6 different compounds at 10 μM concentration. Compounds dissolved in ethanol were further diluted 1:1 with DMSO for dispensing accuracy. DMSO concentration was normalized to 0.3% DMSO to maintain accurate dispensing and account for higher order combinations of multiple compounds. DMSO controls were present on each plate to assess for extent of plate-to-plate variability within a given biological replicate. Upset plots were

generated in R (Bioconductor) to visualize pesticide combinations on the x-axis (https://jokergoo.github.io/ComplexHeatmap-reference/book/upset-plot.html#upset-making-the-plot). Treatment timeline was identical to that described above for toxicant live imaging assays. Toxicants were combined as described. Synergy plots were created from an upset plot of each toxicant combination (ggupset(0.3.0)) and a bar plot of the day 11 neurons' mean THtdTomato brightness (ggplot2 (3.3.5) with a viridis plasma palette (0.5.1). Each combination of two toxicant's THtdTomato count values was compared via Student's *t*-test, and *p*-values were adjusted for multiple testing with Benjamini-Hochberg false discovery rate to *q*-values (reported in Supplementary Data 14).

## Agilent seahorse XF cell Mito stress assay

Cellular respiration of SNCA triplication THtd differentiated neurons in presence of ziram (45708 Milipore Sigma, CAS No 137-30-4) and trifluralin (111020171, MiliporeSigma, CAS No 1582-09-8) was accessed using the Seahorse XF 96 Extracellular Flux Analyzer and the XF Cell Mito stress test kit (Agilent Technologies) following Agilent Technologies guidelines. Mixed dissociated THtd neuronal cultures at day 35 of differentiation were seeded at a volume of 100 μL and a density of $1 \times 10^5$ cells per well in the inner 60 wells of a Polyethylenimine (PEI)-laminin pre-coated Seahorse 96 well plate. The day of the experiment, 20 days after seeding (55 days of total differentiation), cells were treated with 0.3% DMSO (control wells) and differentiation medium containing trifluralin at 90 μM, 60 μM and 30 μM (treatment wells) for 6 h. After this treatment period, medium was removed and 100 μL of the seahorse assay medium was added to each well of the entire plate to start the Mito stress assay, where Oligomycin (1 μM), FCCP (0.5 μM) and Rotenone/Antimycin-A (1 μM) are added at specific time points of the assay to measure metabolic outcome for each condition. After the assay finished, the assay medium was removed from the wells and cells were flash-frozen for subsequent measurement of total protein from each well using Pierce BCA assay kit (23225, Thermo Fisher). Total protein content was used to normalize the data. Data analysis performed with Wave software (Agilent Technologies, version 2.6.1.53). At least three biological replicates (each with its own corresponding set of six technical replicates) were performed for the assay. Distinct and separate differentiations from the same parental iPSC lines were assessed as biological replicates.

## Mitochondrial subunit assays

The mitochondria subunit assay is based on published methodology[35]. Cell pellet generation: SNCA triplication differentiated neurons were dissociated from EBs at day 35 of differentiation. Mixed dissociated neuronal cultures were seeded at a density of $2 \times 10^6$ cells per well in the inner 8 wells of 24 well plates coated with polyethylenimine-Laminin (PEI-Lam). Cells were treated with trifluralin 30 days after seeding (65 days of total differentiation). Differentiation medium containing trifluralin at 90 μM, 60 μM 30 μM, and 10 μM and matched 0.3% DMSO control were prepared. The existing differentiation medium for each well to be treated was replaced with 1 mL of freshly prepared trifluralin-containing differentiation medium. Cells under those conditions were incubated for 6 h. After the incubation period, cells were harvested by pipetting, transferred to 1.5 mL Eppendorf tubes and centrifuged at 4 °C for 5 min at 300 x *g*, medium was removed, and cell pellets were stored at −80 °C for biochemical analysis. Protein Extraction: Aqueous extraction of proteins from cell pellets was done by resuspending the pellets in 100 μL of 1X dilution from 4X blue LDS buffer (B0007, life technologies) plus protease (11697498001, Sigma) and phosphatase (4906837001, Sigma) inhibitors cocktail, sonication for 2x using a tip sonicator at 40% power for 15 s on ice. Samples were boiled for 5 min at 100 °C and then centrifuged for 10 min at 850 x *g* at 4 °C. Proteins contained in supernatant were quantified using the Pierce BCA assay kit (23225, Thermo Fisher). Western blot: Protein

samples (30 μg) were prepared using 4x Bolt LDS sample buffer (B0007, Thermo Fisher) and 10X Bolt Sample reducing agent (B0009, Thermo Fisher) and boiled for 5 min and then loaded into precast Bolt 12-well mini Bis-Tris 4–12% gels and run in MES/SDS buffer for 30 min at 200 V. iBlot dry blotting system (Thermo Fisher) was used to transfer into iBlot nitrocellulose membrane (Thermo Fisher). Licor Odyssey Buffer PBS (Cat. no. 927–40000, Licor) was used to block the membranes at room temperature for 1 h, followed by overnight incubation with agitation at 4 °C with the following primary antibodies: mitoProfile total OXPHOS Human WB antibody cocktail (1:1,000, ab110411), SDHA monoclonal antibody (1:10,000, 459200), anti-TOMM20 (1:1,000, HPA011562) and actin antibody (1:1,200, Sigma A2066). Four 5 min washes with PBS-T (0.05% tween) were performed followed by incubation for 1 h at room temperature with secondary antibodies 680-anti-rabbit, 800-anti-mouse at 1:10,000 dilution in Licor Odyssey plus 0.1% Tween. Membranes were scanned using Licor Clx scanner. Analysis of protein bands was performed using Image Studio software. Three independent differentiations were used for three biological replicates.

### Statistics and reproducibility

A detailed description of analytic methods for the epidemiologic study, described in four parts, can be found in the Supplement Materials, including methods used 1) to describe the extent of agricultural pesticide application in the study area; 2) for the pesticide-wide association analysis, which made use of meta-analyses and logistic regression; 3) for pesticide group overrepresentation analysis; and 4) for PD-associated pesticide clustering. All epidemiologic analyses were done in R version 4.1.0. Statistical analysis for in vitro assays was performed in Graph Pad PRISM software. Power calculations were used to assess feasibility of the pesticide-wide association study, based on 80% power, given the epidemiologic study sample size, and alpha=1.7e-4 to account for multiple testing with a Bonferroni correction. With respect to in vitro assays, no statistical method was used to predetermine sample size. Previous experience with live imaging assays and neuronal survival assays was used to guide approximate numbers of cells needed and fields imaged for each condition. No data were excluded from the analyses. While the in vitro experiments were not randomized, an image analysis algorithm was applied equally to all wells following optimization of parameters on negative control and positive control wells. The investigators were not blinded to allocation during epidemiologic analysis, in vitro experiments, and outcome assessment.

### Reporting summary

Further information on research design is available in the Nature Portfolio Reporting Summary linked to this article.

## Data availability

Further data that was analyzed during the current study cannot be made publicly available due to participant privacy as they provide information about residential location. Deidentified data that support the findings of this study are available on request from the corresponding author BR. Source data are provided with this paper.

## Code availability

Processing code for the source data is provided on github (https://github.com/KCPaul-lab/NatComms_PWAS).

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

## Acknowledgements

This work was supported by grants from: U.S. Army Medical Research Acquisition Activity (USAMRAA) W81XWH1910696 (B.R., L.R., V.K.).; The Michael J. Fox Foundation for Parkinson's Research (MJFF-001018, K.C.P., B.R.); Nikon Corporation (L.R.); a generous gift to the Harvard Stem Cell Institute from the Elizabeth Miller Fund (L.R); American Academy of Neurology Neuroscience Research Training Scholarship (R.K.); National Institute of Environmental Health Science (grant number 2R01ES010544, B.R.); National Heart Lung and Blood Institute (F32HL154644, A.K. and R01HL151684, R.L.). We thank the NeuroTechnology Studio at Brigham and Women's Hospital for providing Agilent Seahorse XFe96 instrument access and consultation on data acquisition and data analysis and Isabel Lam for assistance with graphics and Illustration.

## Author contributions

PWAS and epidemiologic co-exposure conception, design, and analysis: K.C.P. SNCA triplication knock-in line generation, dopamine neuron experiments: R.K. Mitochondrial assays, Seahorse assays: E.L.M. Transgenic reporter constructs: T.A. Dopamine neuron experiments: J.B. Cardiomyocyte differentiation, contractility analysis, pesticide treatment, toxicity analysis: A.K., E.R., H.P., Y.J.L., R.L., R.K. Statistics, graphing of cotton cluster upset plots: K.H. PWAS analysis interpretation and PUR map: M.F., Y.Y. Ambient pesticide exposure assessment: M.C., L.K.T. Neurology exams, interpretation, and data collection for PEG study: J.B., B.R. Experimental design, data analysis, manuscript writing: K.C.P., R.K., E.L.M., L.R., V.K., B.R.

## Competing interests

L.L.R. is a founder of and a member of the Scientific Advisory Board of Vesalius Therapeutics, a private biotechnology company, and an owner of stock options. He is a member of the Scientific Advisory Board of Yumanity Therapeutics and a shareholder. Both companies study Parkinson's disease. B.R., M.C., and R.C.K. have been retained as expert consultants for plaintiffs in a lawsuit on the role of paraquat in Parkinson's disease causation. V.K. is a co-founder of and senior advisor to DaCapo Brainscience and Yumanity Therapeutics, companies focused on central nervous system diseases. The remaining authors declare no competing interests.
