## [Peer Review File · Nature Communications]

A pesticide and iPSC dopaminergic neuron screen identifies and classifies Parkinson-relevant pesticidesREVIEWER COMMENTS

Reviewer #1 (Remarks to the Author):

This is a very interesting and important study highlighting the association between specific pesticides and Parkinson's disease using both epidemiological data and iPSCs. The epidemiological methods and statistical analyses are robust and rigorous, and the findings are a very important addition to the literature. I only have a few comments regarding the overall study design.

- 1. The supplement cover includes tables but only the supplemental figures are shown.**
- 2. For PEG2, were there differences between the 183 who completed only the abbreviated interview vs. those that completed the entire interview?**
- 3. The PD cases had a higher proportion of men compared to controls, which is similar to the higher risk of PD among men. Given the similar household exposure, can you assess whether there are sex differences with regards to exposure and PD? I realize there are several other factors involved including differences in occupation by sex, etc. However, it would be interesting to see if there was a sex interaction.**
- 4. I may have missed it, but how were occupation considered?**
- 5. The patient with iPSC lines derived – please state male or female.**

Reviewer #2 (Remarks to the Author):

The authors developed a ground-breaking field-to-bench approach to identify PD-relevant pesticides. They first investigated exposure to ~ 300 pesticides and PD risk by analysing agricultural pesticide-application records in California to undertake a comprehensive pesticide-wide association study (PWAS). They then assessed the lead pesticides for their effect on dopaminergic neurons and identified 10 toxins which are directly toxic to these neurons. In addition, they identified "cotton-cluster" pesticides - co-exposure to these resulted in markedly greater toxicity than any single pesticide.

Comments: This is an outstanding study. It is by far the best epidemiological study on PD I've seen for a long time. It combines a unique data source (the Californian pesticide application records) with an extremely smart overall study design which includes careful experimental validation of the top epidemiological hits.

I only have a few minor comments:

- 1. Both the introduction and the discussion are a little too long and also a little too self-congratulatory. I suggest the authors shorten both and tone it down a bit.**
- 2. P 8, first line, second paragraph: The authors should explain a little more how "overrepresentation analysis (ORA)" works and justify it in more detail rather than just stating that ORA is also commonly applied to evaluate gene-set overrepresentation.**

Reviewer #3 (Remarks to the Author):

Paul, Krolewski et al performed a screening of pesticides for association with Parkinson's disease and an in vitro toxicity evaluation of candidate hits in induced pluripotent stem cell-derived midbrain dopaminergic (mDA) neurons carrying a-synuclein (SNCA) triplication.

The topic is relevant and overall the data are of interest to the field. However, several issues should be addressed.

- The study is limited by the fact that only one SNCA triplication iPSC line has been used for the in vitro testing experiments. The authors claim that the SNCA triplication genetic background mimics the accumulation of wild-type α -syn occurring in sporadic late-onset PD. However, SNCA triplication causes a very rare Mendelian form of PD and SNCA

baseline levels per se may influence the susceptibility to pesticides. As exposure to pesticides is relevant to all forms of sporadic late-onset PD and the toxic action of pesticides may be modulated by polymorphisms, the in vitro screening should be validated in additional iPSC lines (including at least WT and/or sporadic PD mDA neurons).

- Treatment paradigms are not consistent throughout the manuscript, which makes data interpretation difficult. For instance, cell survival was assessed 11 days after treatment, mitochondrial function was assessed in 65 DIV mDA neurons 6 hours after treatment, mitochondrial protein content was assessed in 65 DIV neurons 24 hrs after treatment. Based on the experimental design depicted in Supplementary Figure 4, treatments for cell imaging assays were performed in 42-48 DIV neurons.

- Cell imaging was performed before treatment, 7 days, and 11 days after treatment. However, the authors only provide results related to last timepoint. All datapoints should be provided.

- Figure 4 and subsequent survival assays: the authors provide data relative to the % of TH+ neurons. Were they normalized on the % of TH+ neurons at baseline? Toxicity should be further validated by additional assays, e.g. LDH or MTT assays.

- The authors should further characterize α -synuclein pathology. For instance, cytoplasmic inclusions containing α -synuclein, pathological forms of α -synuclein, immunoreactivity for ubiquitin.

- Do the selected pesticides, alone or in combination, affect α -syn levels?

- Is there a relationship between pesticide exposure and α -synuclein pathology? Are there mechanisms by which pesticide- α -synuclein interactions could promote mDA neuron death?

Concerning the mechanism of action, do the authors have any hypothesis? The doses employed in the in vitro screening are definitely high (as stated by the authors). Do lower doses affect protein clearance pathways, such as proteasome activity and autophagy? These mechanisms, which are also shared by other pesticides such as paraquat, may be relevant even in the absence of overt neurodegeneration.

- Is there a cell-type specific vulnerability to the selected pesticides? To this end, it would be valuable to test their effect in other neuronal cell types.

- Given that human pesticide toxicity rarely involves exposures to one single agent, the experiment addressing co-exposures is a key experiment with particular translational relevance. Only one co-exposure appears to be significant, namely trifluralin+tribufos. One would expect more combinations of toxicants to be relevant to PD risk. The impact of trifluralin on mitochondrial function has already been described in other mammalian cell types and plants (PMID: 31973855, PMID: 24306369). Furthermore, it has been described that trifluralin accelerates the fibrillation of α -synuclein in vitro (PMID: 12428725). It would be interesting to further characterize the mechanisms of actions of these agents in mDA neurons.

- The authors show that trifluralin alone at 10 μ M produced a 32% decrease in mDA neurons. However, at this concentration no impact on mitochondrial respiration was observed. A chronic low-dose treatment would be more relevant. Furthermore, these data would suggest that the effect of trifluralin on mDA neurodegeneration may be linked to other mechanisms other than mitochondrial dysfunction.

- Figure 7: the authors should measure complex activity, mitochondrial membrane potential, and mitochondrial ROS production.

Response to Reviewers:

We thank the reviewers for their careful reading of the manuscript and thoughtful suggestions. Please find our point-by-point responses to the comments below.

Reviewer #1 (Remarks to the Author):

This is a very interesting and important study highlighting the association between specific pesticides and Parkinson's disease using both epidemiological data and IPSCs. The epidemiological methods and statistical analyses are robust and rigorous, and the findings are a very important addition to the literature. I only have a few comments regarding the overall study design.

1. The supplement cover includes tables but only the supplemental figures are shown.

The supplemental figures are shown in one document ("PWAS NatComms SuppMat.docx"). The supplemental tables, some of which are very large, are described on the document cover page and included as a separate supplemental excel file ("PWAS NatComms SuppTables.xlsx").

2. For PEG2, were there differences between the 183 who completed only the abbreviated interview vs. those that completed the entire interview?

The 183 controls who only completed an abbreviated interview only (due to time constraints as they only agreed to the short version of our interview) were similar to the other controls except for minor differences (age: mean 63.5 (SD=11.6) for the 183 vs 65.6 (11.0) for other controls; 54% male vs 57% male; 50% white vs 60% white). However, by design the two groups differ in length of exposure history as we only asked them to report 3 most recent addresses instead of a complete address history. For the long-term lagged exposure window used in this paper (1974 until 10 years prior to PD or interview for controls), the controls who completed the full interview had on average 25.9 (SD=1.9) years of residential address history while the 183 with the abbreviated history had on average 18.9 (SD=8.8) years of address history and therefore exposure years covered (t-test: $p < 2.2e-16$). For workplace address history, the controls with full interview had on average 20.2 years (SD=8.1), while those with the abbreviated had 10.3 years (SD=10.2) (t-test: $p < 2.2e-16$). This was the primary reason these controls were excluded. We were concerned that pesticide-use trends changed over time. We thus wanted to compare patient and control exposures from the same time periods, both in terms of age ranges and calendar years. This was necessary in order to not underestimate exposures to pesticides primarily applied before 2000 or to overestimate exposure to pesticide primarily applied more recently in the controls. This was also one of the primary reasons for analyzing each study population separately.

That being said, we conducted a sensitivity analysis including the 183 controls with limited exposure data. The results of this are now included in a new supplemental table

(**Supplemental Table 7**), which were mostly robust to including or excluding these controls.

3. The PD cases had a higher proportion of men compared to controls, which is similar to the higher risk of PD among men. Given the similar household exposure, can you assess whether there are sex differences with regards to exposure and PD? I realize there are several other factors involved including differences in occupation by sex, etc. However, it would be interesting to see if there was a sex interaction.

We agree, this would be very interesting to tease out and sex as a biologic variable should be considered here. Further studies will be needed to assess validity and replication because, despite our large sample size of over 1600, we are still underpowered for true interaction analyses. However, for the 68 pesticides implicated in the primary analysis, we now show results stratified by sex in **Supplemental Figure 4**. Most of the pesticides showed similar risk profiles among both men and women. To assess if any variations in risk observed between men and women were statistically different, we ran new analysis including an interaction term in the PD logistic models, as recommended. For most of the pesticides implicated in the PWAS analysis, the risk associations were not meaningfully different between men and women. This can be seen in the **new Supplemental Figure 4**. However, there were 6 pesticides for which we did estimate a pesticide*sex interaction ($p < 0.05$; 10 with $p < 0.10$). In general, the men showed a stronger than expected joint effect. These results were not meaningfully changed when we included or excluded the occupational pesticide indicator. However, it should be noted none of the interactions were significant after multiple testing correction. These interaction results are now also shown in **Supplemental Table 8**.

4. I may have missed it, but how were occupation considered?

We assessed ambient pesticide exposure at workplace address as one of our main exposures. This GIS based exposure assessment method does not estimate occupational exposures from handling pesticides. The analyses we presented did not initially include occupational pesticide exposure. However, we have collected these data and now have conducted and included sensitivity analysis in which we adjust the PWAS results for self-reported occupational use of pesticides or fertilizer (yes/no) as a covariate in the model (**new Supplemental Table 6**). The PWAS results were robust to this covariate adjustment.

5. The patient with iPSC lines derived – please state male or female.

Male – this has now been clarified in the text.

Reviewer #2 (Remarks to the Author):

The authors developed a ground-breaking field-to-bench approach to identify PD-relevant pesticides. They first investigated exposure to ~ 300 pesticides and PD risk by analysing

agricultural pesticide-application records in California to undertake a comprehensive pesticide-wide association study (PWAS). They then assessed the lead pesticides for their effect on dopaminergic neurons and identified 10 toxins which are directly toxic to these neurons. In addition, they identified "cotton-cluster" pesticides - co-exposure to these resulted in markedly greater toxicity than any single pesticide.

Comments: This is an outstanding study. It is by far the best epidemiological study on PD I've seen for a long time. It combines a unique data source (the Californian pesticide application records) with an extremely smart overall study design which includes careful experimental validation of the top epidemiological hits.

I only have a few minor comments:

1. Both the introduction and the discussion are a little too long and also a little too self-congratulatory. I suggest the authors shorten both and tone it down a bit.

We agree. Both sections have been edited and revised per the recommendation.

2. P 8, first line, second paragraph: The authors should explain a little more how "overrepresentation analysis (ORA)" works and justify it in more detail rather than just stating that ORA is also commonly applied to evaluate gene-set overrepresentation.

We agree. This explanation has been elaborated and can now be found in the 'Novel pesticides associated with PD in a pesticide-wide association analysis (PWAS)' section of the manuscript (page 9, paragraph 2).

Reviewer #3 (Remarks to the Author):

Paul, Krolewski et al performed a screening of pesticides for association with Parkinson's disease and an in vitro toxicity evaluation of candidate hits in induced pluripotent stem cell-derived midbrain dopaminergic (mDA) neurons carrying a-synuclein (SNCA) triplication.

The topic is relevant and overall the data are of interest to the field. However, several issues should be addressed.

- The study is limited by the fact that only one SNCA triplication iPSC line has been used for the in vitro testing experiments. The authors claim that the SNCA triplication genetic background mimics the accumulation of wild-type α -syn occurring in sporadic late-onset PD. However, SNCA triplication causes a very rare Mendelian form of PD and SNCA baseline levels per se may influence the susceptibility to pesticides. As exposure to pesticides is relevant to all forms of sporadic late-onset PD and the toxic action of pesticides may be modulated by polymorphisms, the in vitro screening should be validated in additional iPSC lines (including at least WT and/or sporadic PD mDA neurons).

Thank you for this suggestion. We chose the triplication line, as opposed to a mutation, because it models wild-type alpha-synuclein accumulation, thus recapitulating the situation in sporadic disease. Moreover, polymorphisms at the SNCA locus that increase SNCA expression are well-validated GWAS hits predisposing to sporadic late-onset disease. That being said, there is no question that that what the reviewer points out is true – the triplication families have unusually early-onset and aggressive synucleinopathy and the baseline levels of alpha-synuclein in the SNCA triplication iPSC line could theoretically have influenced susceptibility to pesticides. To address this concern, we generated a new iPSC line with THtdTomato knock-in. The line originally harbored an alpha-synuclein point mutation (E46K). We introduced the TH reporter and then genetically corrected it to create a two-copy SNCA wild-type line, as specified by the reviewer. We chose a genetically corrected line from a PD patient to simulate a “PD permissive” genetic background. We tested the ten toxic PEG pesticides in this line. All ten pesticides were validated as toxic in this line. Data are presented in the new **Supplementary Figure 10**.

- Treatment paradigms are not consistent throughout the manuscript, which makes data interpretation difficult. For instance, cell survival was assessed 11 days after treatment, mitochondrial function was assessed in 65 DIV mDAN neurons 6 hours after treatment, mitochondrial protein content was assessed in 65 DIV neurons 24 hrs after treatment. Based on the experimental design depicted in Supplementary Figure 4, treatments for cell imaging assays were performed in 42-48 DIV neurons.

The reviewer makes an important point, and we appreciate the confusion here. This point has now been clarified in the text more carefully. We endeavored to keep assay conditions similar whenever possible. However, each assay reflects a different biological process and have different sensitivities – depending on the biological readout, the timing will thus be different. This is why assay conditions required different timing of treatments for the live imaging, mitochondrial function, and mitochondrial protein content assays, respectively. Some practical constraints also came into play – live imaging assays were performed earlier in the culture period due to the tendency of mDA neurons to clump with prolonged time in 2D culture. Clumping complicates live imaging quantification and reduces accuracy of the measurements but does not interfere with biochemical measures. Mitochondrial function was tested at early time points (6 hours after treatment) with a goal of targeting more direct metabolic effects from the pesticide treatment (expected to occur at shorter time points compared to overt toxicity). We also wanted to avoid prolonged exposure that could lead to nonspecific mitochondrial damage or dysfunction to accumulate. Mitochondrial protein content, on the other hand, was assayed to measure steady state changes that could result from exposure, and thus assayed at a later time point. We followed literature precedent for these assays (e.g. Monzio Compagnoni G, et al Stem Cell Reports 2018).

- Cell imaging was performed before treatment, 7 days, and 11 days after treatment. However, the authors only provide results related to last timepoint. All datapoints should be provided.

We agree. Requested data are now provided in **Supplementary Figure 7B and C**.

- Figure 4 and subsequent survival assays: the authors provide data relative to the % of TH+

neurons. Were they normalized on the % of TH+ neurons at baseline? Toxicity should be further validated by additional assays, e.g. LDH or MTT assays.

We apologize for any confusion about the survival assay readout. To clarify:

- Actual numbers of counted THtdT neurons are presented. Each dot is the raw number of neurons counted from four fields of one well of a 384well plate. Data are not a ratio or percentage.
- Raw data rather than average data were provided to maximize data transparency and allow readers to visualize the data spread and variability.
- In accordance with reviewer request, we now obtained multiple independent measurements through additional analysis and additional experiments to provide orthogonal measures of toxicity. These data are presented in **Supplementary Figures 7 and 9**:
 - Total neurite length was measured as a distinct readout of cell health and toxicity (**Supplementary Figure 7A**).
 - Cell shrinkage often precedes and accompanies cell death (Bortner PMID 12821680). Therefore, cell size was estimated using the area of THtdT fluorescence per cell (**Supplementary Figure 7D**).
 - Intensity of THtdT signal was also independently measured and plotted for all pesticides (**Supplementary Figure 7E**).

Top-hit pesticides were also assessed for viability using a CellTitreGlo assay under the same experimental conditions as in Figure 4 to eliminate any influence of the THtdT reporter on results for viability. These data confirmed reduction of viable cells as expected with concentrations at or below the screening concentration (**Supplementary Figure 9**).

- The authors should further characterize α -synuclein pathology. For instance, cytoplasmic inclusions containing α -synuclein, pathological forms of α -synuclein, immunoreactivity for ubiquitin.

These are excellent suggestions that we are pursuing in detail for a follow up manuscript. Rigorous examination of these questions will require additional cellular reagents that we have in development and plan to utilize to address these questions. In general, we find that endogenous levels of alpha-synuclein in iPSC neurons cultured in our conditions is low – certainly far lower than what we see in human brain tissue. Nevertheless, it is detectable. For the toxicant we focused on, trifluralin, we assessed phosphorylation of alpha-synuclein at Ser 129 by Western Blot. This modification is the most commonly used marker for pathologic alpha-synuclein and is compared to total alpha-synuclein levels. A representative Western Blot is shown in **Supplementary Figure 13**. It indicates that neither pS129 or total alpha-synuclein levels changed appreciably with trifluralin exposure.

- Do the selected pesticides, alone or in combination, affect a-syn levels?
- Is there a relationship between pesticide exposure and α -synuclein pathology? Are there

mechanisms by which pesticide- α -synuclein interactions could promote mDA neuron death? Concerning the mechanism of action, do the authors have any hypothesis? The doses employed in the in vitro screening are definitely high (as stated by the authors). Do lower doses affect protein clearance pathways, such as proteasome activity and autophagy? These mechanisms, which are also shared by other pesticides such as paraquat, may be relevant even in the absence of overt neurodegeneration.

These are excellent suggestions. Our main objective in the current study is to couple quantitative epidemiology with quantitative cell biology and highlight a platform to flag specific toxicants that are likely to be directly toxic to dopaminergic neurons. These will be prioritized for more detailed mechanistic studies, such as those the reviewer suggests. As noted in response to point 5, we are generating a number of additional reagents to test these ideas thoroughly, including reporter and knock-in cell lines. We believe those types of experiments are beyond the scope of the current study.

- Is there a cell-type specific vulnerability to the selected pesticides? To this end, it would be valuable to test their effect in other neuronal cell types.

We agree with this excellent suggestion. We enlisted additional collaborators to generate cardiomyocytes from the SNCA triplication iPSC line. Testing pesticides toxic to mDA neurons revealed that only a subset were also toxic to cardiomyocytes as determined by cell counts. These data are presented in **Supplementary Figure 11** and imply specificity, even at the relatively high doses we used.

- Given that human pesticide toxicity rarely involves exposures to one single agent, the experiment addressing co-exposures is a key experiment with particular translational relevance. Only one co-exposure appears to be significant, namely trifluralin+tribufos. One would expect more combinations of toxicants to be relevant to PD risk. The impact of trifluralin on mitochondrial function has already been described in other mammalian cell types and plants (PMID: 31973855, PMID: 24306369). Furthermore, it has been described that trifluralin accelerates the fibrillation of α -synuclein in vitro (PMID: 12428725). It would be interesting to further characterize the mechanisms of actions of these agents in mDA neurons.

Thank you for pointing out the α -synuclein fibrillation reference. We have included that in the revised manuscript for additional context and discussed this in the text. We are very interested in mechanisms of action for these pesticides in mDA neurons. Those experiments will be the focus of a follow up manuscript with additional reagents, including transgenic reporters that read out alpha-synuclein aggregation. We do however show that in our current testing conditions trifluralin does not alter alpha-synuclein phosphorylation at Ser129 (see response to point 5). In our experience this is a post-translational modification that always accompanies alpha-synuclein aggregation, whether in fibril-rich or membrane-rich inclusions (we extensively characterized this in a recent preprint Lam et al., bioRxiv doi: <https://doi.org/10.1101/2022.11.08.515615>).

- The authors show that trifluralin alone at 10 μ M produced a 32% decrease in mDA neurons.

However, at this concentration no impact on mitochondrial respiration was observed. A chronic low-dose treatment would be more relevant. Furthermore, these data would suggest that the effect of trifluralin on mDA neurodegeneration may be linked to other mechanisms other than mitochondrial dysfunction.

We agree with the reviewer that dosing and timing are challenging aspects of the *in vitro* model. Given the complexity of proposed mechanisms of action for trifluralin (microtubules, mitochondria, alpha synuclein effects), we are pursuing more sensitive assays that will help clarify the relative importance of the different proposed mechanisms. We now make reference to this in the discussion.

- Figure 7: the authors should measure complex activity, mitochondrial membrane potential, and mitochondrial ROS production.

These are excellent suggestions that we are pursuing in detail for a follow up manuscript.

REVIEWERS' COMMENTS

Reviewer #1 (Remarks to the Author):

The authors have addressed all of my concerns. This is an important, rigorous paper.

Reviewer #2 (Remarks to the Author):

no further concerns

Reviewer #3 (Remarks to the Author):

The authors addressed our concerns with additional experiments and/or gave the rationale for a follow-up work. Given the importance and completeness of the current results, we believe that the manuscript is suitable for publication.

Response to Reviewers:

REVIEWERS' COMMENTS

Reviewer #1 (Remarks to the Author):

The authors have addressed all of my concerns. This is an important, rigorous paper.

-No response warranted. Thank you.

Reviewer #2 (Remarks to the Author):

no further concerns

-No response warranted. Thank you.

Reviewer #3 (Remarks to the Author):

The authors addressed our concerns with additional experiments and/or gave the rationale for a follow-up work. Given the importance and completeness of the current results, we believe that the manuscript is suitable for publication.

-No response warranted. Thank you.